# Emerging Role of Autophagy in Governing Cellular Dormancy, Metabolic Functions, and Therapeutic Responses of Cancer Stem Cells

**DOI:** 10.3390/cells13050447

**Published:** 2024-03-04

**Authors:** Meenakshi Tiwari, Pransu Srivastava, Sabiya Abbas, Janani Jegatheesan, Ashish Ranjan, Sadhana Sharma, Ved Prakash Maurya, Ajit Kumar Saxena, Lokendra Kumar Sharma

**Affiliations:** 1Department of Biochemistry, All India Institute of Medical Science, Patna 801507, India; 2Department of Molecular Medicine & Biotechnology, Sanjay Gandhi Post Graduate Institute of Medical Science, Lucknow 226014, India; 3Department of Neurosurgery, Sanjay Gandhi Post Graduate Institute of Medical Sciences, Lucknow 226014, India; 4Department of Pathology/Lab Medicine, All India Institute of Medical Science, Patna 801507, India

**Keywords:** cancer stem cells, autophagy, metabolic functions, chemotherapy, radiotherapy, mitophagy, quiescence

## Abstract

Tumors are composed of heterogeneous populations of dysregulated cells that grow in specialized niches that support their growth and maintain their properties. Tumor heterogeneity and metastasis are among the major hindrances that exist while treating cancer patients, leading to poor clinical outcomes. Although the factors that determine tumor complexity remain largely unknown, several genotypic and phenotypic changes, including DNA mutations and metabolic reprograming provide cancer cells with a survival advantage over host cells and resistance to therapeutics. Furthermore, the presence of a specific population of cells within the tumor mass, commonly known as cancer stem cells (CSCs), is thought to initiate tumor formation, maintenance, resistance, and recurrence. Therefore, these CSCs have been investigated in detail recently as potential targets to treat cancer and prevent recurrence. Understanding the molecular mechanisms involved in CSC proliferation, self-renewal, and dormancy may provide important clues for developing effective therapeutic strategies. Autophagy, a catabolic process, has long been recognized to regulate various physiological and pathological processes. In addition to regulating cancer cells, recent studies have identified a critical role for autophagy in regulating CSC functions. Autophagy is activated under various adverse conditions and promotes cellular maintenance, survival, and even cell death. Thus, it is intriguing to address whether autophagy promotes or inhibits CSC functions and whether autophagy modulation can be used to regulate CSC functions, either alone or in combination. This review describes the roles of autophagy in the regulation of metabolic functions, proliferation and quiescence of CSCs, and its role during therapeutic stress. The review further highlights the autophagy-associated pathways that could be used to regulate CSCs. Overall, the present review will help to rationalize various translational approaches that involve autophagy-mediated modulation of CSCs in controlling cancer progression, metastasis, and recurrence.

## 1. Introduction

Cancer remains the second leading cause of death worldwide [1]. Despite significant efforts from researchers and clinicians, cancer is still considered a death sentence, which is even true for high-grade malignancies. High mortality among cancer patients largely occurs because of tumors’ resistance to existing therapies, which leads to recurrence. To develop effective therapies, it is important to understand the process of tumorigenesis and the specific molecular pathways that need to be targeted to overcome therapeutic resistance.

Based on extensive research, it has been identified that the majority of tumors are derived from cancer stem cells (CSCs), which are transformed from normal stem cells (SCs) or differentiated cells due to various genetic or epigenetic alterations [2]. In the late 1990s, John Edgar Dick’s group pioneered the discovery of cancer stem cells (CSCs) by proposing a leukemia-initiating cell hierarchy model. This model suggested that human acute myeloid leukemia (AML) follows a hierarchical organization originating from primitive hematopoietic cells [3].

Subsequently, similar populations of CSCs were identified in various solid tumors, spanning breast, brain, prostate, ovarian, gastric, lung, and pancreatic cancers [4,5,6,7]. These CSCs were identified as a rare population of primitive cells that were long lived, could exist in proliferative or quiescent/dormant stages, were apoptosis resistant, multipotent, and most importantly possessed self-renewal capacity. This led to the proposal that CSCs are the seed of cancer and are responsible for tumor initiation, progression, and cellular heterogeneity, resistance to therapies, recurrence, and metastasis. However, the molecular mechanisms determining the proliferation/differentiation, chemoresistance, and other characteristics of CSCs are highly complex and remain poorly defined [8].

Autophagy is a catabolic process that involves the degradation of dysfunctional or unwanted cellular components through cellular lysosomal machinery. Autophagy serves to maintain cellular homeostasis under physiological conditions and is also activated under various pathological conditions and determines the outcome of the disease. The role of autophagy in CSC maintenance, proliferation, differentiation, and resistance to therapies is emerging [9,10,11,12,13]. Thus, autophagy is considered a potential therapeutic target in various cancers to eliminate CSCs. The present review explores recent advances in understanding the role of autophagy in CSCs with a special focus on various molecular mechanisms involved, which can act as potential therapeutic targets in otherwise therapy-resistant CSCs. The present review covers the basic mechanisms of CSC biology and sheds light on the relationship between autophagy and CSCs via the regulation of cellular dormancy, metabolic pathways, mitochondrial regulation, and therapeutic response. Understanding autophagy-related biological targets will contribute to the design of innovative therapeutic strategies aimed at the selective elimination of CSCs or induction of their differentiation.

## 2. CSCs: Seed of Tumors

According to the American Association of Cancer Research, a CSC is defined as “a cell within a tumor that has the capacity to self-renew and to cause the heterogeneous lineage of cancer cells within a tumor” [14]. These CSCs account for a rare subset (~0.1–2%), thought to have originated from the pool of normal SCs or differentiated cells [12,15,16]. Normal SCs have a very strict machinery to control their division rate; however, mutations and epigenetic changes in these regulatory pathways lead to the generation of CSCs, which gain the capacity of uncontrolled proliferation leading to tumorigenesis [17]. Thus, it is important to understand the properties of CSCs to prevent tumorigenesis and eradicate them along with bulk tumor cells.

### 2.1. Understanding the Origin of CSCs

It is well documented that CSCs are initiated from normal tissue SCs or differentiated cells after various genetic and epigenetic modifications [18]. It is also postulated that CSCs can arise due to fusion between cancer cells and adult SCs or via the process of transformation during inflammation, infection, or metabolic processes that could cause mutations. Epigenetic changes, such as aberrant DNA methylation that involve hypomethylation or hypermethylation of CpG islands and histone modification, are also prominent features of carcinogenesis [19,20]. Supporting this hypothesis, studies have revealed that long-lived cells are prone to cancer-causing mutations [21,22]. The hypothesis is that cellular reprograming induces the expression of pluripotency factors in differentiated cells, leading to aberrant stemness and cancer [19,23]. Furthermore, CSC plasticity is maintained by various factors, including genetics, epigenetics, and the microenvironment of tumors. It is interesting to note that reports have supported the CSC plasticity model where transitions take place between CSCs and non-CSCs. Friedmann-Morvinski et al. identified that even the highest differentiated cells, neurons, under the influence of genetic and epigenetic modulation can dedifferentiate and may lead to tumor formation with heterogeneous populations as seen in malignant gliomas [24]. Similarly, in the case of intestinal adenomas, dedifferentiation leading to transformation from non-tumorigenic to tumorigenic CSCs has been identified.

Taken together, as depicted in Figure 1, various models have been proposed for the origin of CSCs that mainly include: (1) the stochastic or clonal evolution (CE) model, in which any cell that accumulates mutations may have tumorigenic potential, (2) the hierarchy or CSC model, suggesting tumorigenic potential of CSC tumors unlike the other cells, and (3) the plasticity model, according to which any cancer cell has the potential to convert into a CSC and vice versa [1,12,25]. It is suggested that CSCs are the seed cells of tumors that are responsible for tumor initiation, recurrence, and resistance to chemotherapy. These CSCs are autonomous and are not as strictly controlled by the local microenvironment as normal SCs. Furthermore, due to the lack of maintenance of genomic integrity, such cells may further accumulate mutations and acquire the ability to differentiate into multiple tumor cell types, which may further contribute to tumor heterogeneity and aggressiveness [26,27].

### 2.2. Markers of CSCs

Similar to normal SCs, CSCs are unspecialized cells that can proliferate and divide, renew themselves for long periods, and give rise to specialized cells. These cells show a higher expression of stem-cell marker genes (i.e., cluster of differentiation (CD)34, CD44, CD123, CD133, octamer-binding transcription factor ¾ (Oct3/4), SRY-Box transcription factor 2 (Sox2), Nanog, c-Kit, leukemia inhibitory factor (Lif), adenosine 5′-triphosphate (ATP) binding cassette subfamily G member 2 (ABCG2), aldehyde dehydrogenase (ALDH), and C-X-C motif chemokine receptor 1 (CXCR1)) and contribute to tumor formation as well as progression [15,26,28]. Based on the expression of these stem-cell-specific markers, identification and characterization of CSCs in malignant tissue is possible and can provide insight into the pathogenic mechanisms, interaction of CSCs with the tumor microenvironment, and therapeutic targeting.

### 2.3. Signaling Pathways That Drive CSCs

Multiple signaling pathways that regulate normal SCs also regulate the self-renewal, proliferation, survival, and differentiation properties of CSCs [29]. These pathways include the Wnt/β-catenin, Notch, Janus kinase (JAK)/signal transducer and activator of transcription (STAT), Hedgehog, phosphoinositide 3-kinase (PI3K)/phosphatase/Akt/mTOR (mTORC1 and mTORC2), phosphatase and tensin homolog (PTEN), nuclear factor κ-B (NF-κB), maternal embryonic leucine zipper kinase (MELK), TGF-β, STAT, and Hippo-YAP/TAZ pathways [29,30,31]. Under normal conditions, signaling pathways that regulate the functions and properties of normal SCs are highly coordinated and controlled, whereas in CSCs, these pathways are dysregulated (either upregulated or down regulated) thereby promoting the tumorigenic potential of these cells [18]. Complex interactions involving inter-pathway crosstalk further determine regulator outcomes in CSC.

In addition to genetic factors, CSCs are also controlled by epigenetic factors such as DNA methylation, chromatin remodeling, and non-coding RNA [32]. Studies have demonstrated that the upregulation of zeste homolog 2 (EZH2), which is the catalytic subunit of polycomb repressive complex 2 (PRC2) and has histone methyltransferase activity, plays a crucial role in the expansion and maintenance of CSCs [33]. Overexpression of histone deacetylases (HDACs) 1, 6, 7, and 8 leads to deacetylation of transcription factors and other cellular proteins in CSCs [34]. Additionally, epigenetic mechanisms control the expression of ABCG2, a drug efflux pump crucial for chemoresistance, in CSCs [35]. Furthermore, epigenetic modulations promote epithelial–mesenchymal transition and therapeutic resistance in CSCs [36]. These epigenetic modifications crosstalk with genetic and post-translational mechanisms and regulate CSC properties. Moreover, because of the impact of environmental pollutants and carcinogens such as polycyclic aromatic hydrocarbons (PAHs) on epigenetic modifications, a link has been proposed between genetic, epigenetic, and environmental factors in determining the fate of CSCs [37].

The crucial role of the tumor microenvironment has also been demonstrated in regulating the properties of CSCs. Reports have suggested that tissue hypoxia leads to the activation of hypoxia-inducible factors (HIFs), which results in the enrichment of CSCs via the process of dedifferentiation in various cancers, including breast CSCs (BSCs), glioblastoma CSCs (GSCs), and other solid tumors [38].

### 2.4. Why Do We Need a Deeper Understanding of CSC Regulation?

The current notion strongly advocates CSCs as the major culprit of tumorigenesis and tumor relapse, as these cells are therapies-resistant [39]. Thus, it is imperative to address several key questions to therapeutically target these cells and achieve better treatment outcomes. The key questions include: What are the markers that are specific to these cells?; What are the differences in the signaling pathways that drive them?; What are the epigenetic alterations and how do these cells undergo reprograming and develop plasticity?; What are the metabolic pathways that drive these cells?; Why are these cells therapies-resistant and how can we make them sensitive to existing therapies?; What drives the quiescence of these cells and how do they reenter the proliferative stage that leads to tumor regrowth?

Apart from the ability of CSCs to become dormant and evade antineoplastic drugs, several mechanisms are utilized by these cells to protect themselves from therapy. CSCs have increased drug efflux capacities due to the overexpression of multi-drug resistance (MDR) transporters, enhanced DNA repair capacity, and resistance to apoptosis [40]. Therefore, developing an understanding of the regulatory pathways governing CSC functions is crucial for their eradication. In this regard, the role of autophagy is emerging as a crucial regulator of CSC functions. The role of autophagy in cancer is highly complex, and its differential functions strictly depend on multiple factors such as the stage of cancer, type, growth, and progression [41]. It is now established that autophagy is dysregulated in CSCs and plays a role in therapeutic resistance, as depicted in Figure 2. However, before employing these pathways for therapeutic purposes, a deeper understanding should be developed for harnessing the maximum potential of these pathways. In the following sections, we have discussed the recent understanding of autophagy and its role in the regulation of CSCs and its potential as a therapeutic target.

## 3. Autophagy

Autophagy is a conserved catabolic cellular process for the recycling and removal of damaged or unwanted cellular components. It involves the formation of double-membrane structures called autophagosomes that fuse with lysosomes for degradation [42,43]. The role of autophagy as a physiological process is well established and is activated in response to cellular stressors such as nutrient deficit and oxidative stress, which regulate diverse cellular functions including growth, differentiation, cell death, and macromolecule and organelle turnover. Apart from physiological processes, it plays a crucial role in a variety of pathological conditions, including cancer, aging, and pathogen response [44]. Based on the substrates and regulatory proteins, three types of mammalian autophagy have been defined, as depicted in Figure 3.

(1)Macroautophagy—This is the most common and well-studied form of autophagy that involves bulk degradation of cellular components and organelles via engulfment in double-membrane vesicles called autophagosomes, which fuse with lysosomes for degradation. This is discussed in detail in the following sections.(2)Microautophagy—In this type of autophagy, the cytosolic portion targeted for degradation is directly engulfed by lysosomes for degradation by hydrolases. It is a random process that involves various steps that mainly include membrane invagination, vesical formation and elongation, and degradation and recycling of vesicles by lysosomes, followed by the release of nutrients by Atg22p [45]. This process promotes cell survival under conditions of starvation and nitrogen deprivation and maintains cellular organelle size, composition, and growth.(3)Chaperone-mediated autophagy (CMA)—This is a more selective degradation process in which soluble cytosolic proteins tagged with a specific pentapeptide motif (KFERQ) are targeted for degradation via lysosomes without the formation of vesicles. Such proteins are recognized by heat shock cognate chaperone of 70 kDa (Hsc70) and transported to the lumen of lysosomes by lysosome-associated membrane protein type 2A (LAMP-2A) and subjected to degradation. CMA is known to play an important role in the selective degradation of proteins related to glycolytic pathways, mutant p53, and others [46,47].

Of all these forms, macroautophagy is the most commonly studied form and is referred to as autophagy in this review unless defined.

### 3.1. Mechanism of Autophagy

The process of autophagy is regulated by the interplay of specific proteins named after their corresponding genes called autophagy-related genes (ATGs). Yoshimori Osumi performed remarkable research in the identification of these regulatory genes and was awarded the prestigious Nobel Prize in Physiology and Medicine in 2016. These ATG proteins regulate autophagy by assembling into functional complexes involved in the autophagic process. In the following section, the mechanism of macroautophagy is discussed, which is referred to as autophagy hereafter.

As depicted in Figure 4, autophagy is initiated under conditions such as deprivation of growth factor and amino acids, which inhibits the mammalian target of rapamycin (mTOR), a central cell-growth regulator that integrates growth factor and nutrient signals and inhibits autophagy. In addition, during bioenergetically stressed conditions when ATP is limited, autophagy is initiated through the energy-sensing pathway via the activation of AMP-activated protein kinase (AMPK). Both the mTOR and AMPK pathways regulate autophagy by phosphorylating ATG1/unk-51-like autophagy activating kinase 1 (ULK1), a serine–threonine kinase. AMPK promotes autophagy by directly activating Ulk1 through the phosphorylation of Ser 317 and Ser 777. In contrast, mTORC1 activity prevents Ulk1 activation by phosphorylating Ulk1 Ser 757 and disrupting the interaction between Ulk1 and AMPK. This coordinated phosphorylation is important for Ulk1 in autophagy induction. Thus, under stress conditions, mTORC1 dissociates from the ATG1/ULK1 complex and thus the level of phosphorylation decreases, enabling the complex to initiate autophagy. mTORC2, which is rapamycin insensitive, can also inhibit autophagy via the activation of the AKT pathway (Figure 4) [48,49].

Once activated, ULK1 forms a complex involving ATG13 and FIP200, also called the pre-initiation complex, which translocates to a discrete location on the ER or other membrane marked by ATG9. Following this, a complex mechanism recruits and activates phosphatidylinositol 3 kinase class III (PI3K III or hVps34). This enzyme together with Beclin-1, UV-RAG, ATG14, and Ambra1, forms a larger assembly known as the initiator complex which is assembled at specific parts of the endoplasmic reticulum called omegasomes. The activation of this initiator complex can be directly controlled by AMPK and blocked by Bcl-2/Bcl-xL.

The initiation process requires activation of PI3K-III, which converts phosphatidylinositol to phosphatidylinositol 3-phosphate (PI3P) [50,51]. Generation of PI3P dictates the site by changing the lipid composition with marked curvature along with recruitment of WIPI2B at which the double membrane subsequently elongates. The elongation process is driven by a ubiquitin-like system involving E1–E2–E3 ligases that regulate the ubiquitination of proteins that are marked for autophagy.

The membrane for autophagosome formation mainly comes from cellular sources such as the ER, mitochondria, recycling endosomes, and plasma membrane [52]. Once the process of phagophore formation is initiated, its elongation takes place by involving two different ubiquitin-like conjugation pathways, both of which are catalyzed by ATG7. A complex involves conjugation of ATG5 and ATG12 involving the E1 ligase ATG7 and E2 ligase ATG10, thus forming the E3 complex ATG16L and ATG5-ATG12. This complex associates with the outer membrane of the phagophore [53,54]. Another step is the formation of LC3-II, which is produced from LC3 by (1) photolytic cleavage of LC3 to LC3I by ATG4 and (2) ATG7 (E1) binding to and activating LC3I, which is then transferred to ATG3 (E2). LC3I is then transferred to the E3 complex ATG5-ATG12-ATG16L, where conversion of LC3I to LC3II/LC3 phosphatidylethanolamine (PE) conjugate takes place. LC3II is recruited on both outer and inner membranes of the autophagosome and is required for the elongation step [55].

As autophagy progresses, the autophagosome membrane expands and engulfs the cytoplasmic components and organelles designated to be degraded. Proteins with ubiquitin binding sites, such as sequestosome1 (SQSTM1)/p62, bind to ubiquitin-tagged proteins. Such proteins are directed to autophagosomes because of their ability to bind to LC3, which facilitates this process by delivering SQSTM1-containing protein aggregates to autophagosomes, which are further translocated to lysosomes to form autophagolysosomes and undergo destruction. After the closure of autophagosomes, which is facilitated by STX17, a protein resident in the endoplasmic reticulum, the mature autophagosomes move towards lysosomes. This movement is driven by motor proteins like dynein and kinesin along microtubules, as well as actin. The tethering of autophagosomes to lysosomes involves homotypic fusion and protein sorting (HOPS), facilitated by the HOPS complex. RAB7 and adaptors facilitate fusion by binding with Q-SNARE and STX17. The fusion of autophagosomes and lysosomes leads to the release of lysosomal content in the space between the autophagosomal membrane. Inner membrane degradation occurs in an LC3II-dependent manner. After breaking down the inner autophagosomal membrane, its contents are degraded into amino acids, glucose, and other building blocks that are released by lysosomal efflux transporters and recycled. These autolysosomes disintegrate once autophagy is terminated [56,57,58].

### 3.2. Autophagy in Cancer: A Complex Role

Autophagy is a well-regulated process that plays a crucial role at different developmental stages, including differentiation, cellular homeostasis, renovation, and many other physiological processes [59]. Dysregulation of autophagy plays a crucial role in a variety of pathological conditions such as infections, neurodegeneration, aging, heart disease, and cancer [47,60]. The relationship between autophagy and cancer is complicated, and whether it plays a tumor-suppressive role or promotes tumor survival remains puzzling [61]. It has been identified that autophagy has different functions depending on various factors such as the stage of cancer and genetic makeup [62]. Observations from mice harboring defects in autophagy genes showed a higher incidence of spontaneous or chemically induced tumors, indicating the role of autophagy as a tumor suppressor. The initial evidence came from the group of Levin who identified that a targeted mutant mouse model with monoallelic deletion of beclin1, yeast autophagy-related gene 6 (Atg6), which is an important regulator of autophagy, promotes tumorigenesis [63,64]. This model was based on observations that the beclin1 gene is monoallelically deleted in 40–75% of cases of human sporadic breast, ovarian, and prostate cancer. Similarly, data from human neoplasms such as breast, ovarian, prostate, and colorectal cancers also indicated that the loss of autophagic genes is associated with tumor development. In recent years, fasting- and caloric-restriction-dependent autophagy induction has been shown to improve anticancer immune surveillance, thereby promoting tumor growth arrest and improvement of chemotherapeutic outcomes [41,65]. It was also observed that p53 plays an important role in regulating the autophagic pathway [66]. In an interesting study, it was identified that mice lacking the essential autophagy genes Atg5 or Atg7 accumulate low-grade, pre-malignant pancreatic intraepithelial neoplasia lesions, but progression is blocked. However, in mice containing oncogenic *Kras* along with loss of p53, autophagy inhibition no longer blocks tumor progression but accelerates tumor onset. Thus, p53 emerged as an important determinant of the role of autophagy in tumor initiation and progression [67].

In addition to other mechanisms, the glycoconjugates plays a significant role in regulating autophagy [68,69]. Importantly, aberrant glycosylation is a characteristic feature of malignancy and is regarded as one of several biomarkers associated with it [70]. An overproduction of glycoproteins and glycolipids has been shown in cancer, which affects the malignancy potential, tumor immune surveillance, and prognosis [70]. Interestingly, glycoconjugates, including glycoproteins and glycolipids, play an essential role in the formation of autophagosomes and lysosomes [68]. Ganglioside, a glycolipid, is involved in the biogenesis of autophagosome membranes and in maintaining autophagic flux. Co-localization of GD3 with PI3P promotes recruitment of the core machinery of autophagosome formation [71].

Autophagy-regulating proteins are also regulated by glycosylation, and oligosaccharide exposure in the cytosol triggers an autophagic response. Extracellular and intracellular lectins (protein-bound carbohydrates) also regulate autophagy induction. Glycan signaling from the extracellular matrix regulates AMPK and mTORC1 activity and thus directly impacts autophagic processes [68,69]. Furthermore, autophagy also regulates glycoconjugate turnover, including the elimination of free glycans/oligosaccharides, and may influence malignancy via such regulation [68].

Because aberrant glycosylation is a hallmark of cancer, understanding the glycobiology of CSCs might provide important insights into CSC biology and may define its role more specifically in various processes such as tumorigenesis and drug resistance [72]. Recent investigations suggest abnormal glycolic profiles of CSCs compared with normal cells and non-CSCs [73]. It is imperative to understand the specific pattern of glycoconjugates along with their biological functions in CSCs, which can be used as biomarkers for disease progression as well as therapeutic targeting. With advances in recent technology involving high-throughput array-based cell analysis along with quantitative mass spectrometry-based glycomics, a more specific pattern of glycoconjugates associated with CSCs can be identified. Moreover, understanding the crosstalk between CSCs and autophagy might provide important clues for therapeutic targeting.

Thus, autophagy is considered a double-edged sword because of its tumor-suppressive or promoting role, which varies in different cancers and depends on multiple factors. The dual role of autophagy can be explained by understanding its pathways in different tumors. For example, basal autophagy prevents tumor initiation by removing damaged and defective macromolecules and organelles, which could otherwise cause an increase in reactive oxygen species (ROS) and lead to compromised genomic integrity tumor initiation. In contrast, during tumor progression, autophagy has been shown to protect tumor cells from cell death by regulating oxidative stress, DNA damage, and hypoxia caused by adverse conditions in which the cancer cells reside as well as therapeutic stress. In such cases, autophagy-mediated recycling of cellular content helps to meet the high metabolic and energy requirements for tumor cells, thus supporting tumor growth and proliferation in otherwise adverse conditions. Therefore, understanding the underlying role of autophagy and its modulation could offer novel therapeutic opportunities to drive cancer cells toward cell death [2,74].

### 3.3. Role of Autophagy in the Regulation of Normal SCs

Before we discuss the role of autophagy in CSCs, it is important to discuss the role of autophagy in normal SCs because CSCs may originate from these cells and exhibit several similar characteristics. The mechanisms by which autophagy contributes to stemness and why SCs are more dependent on autophagy than non-SCs is an area of interest for many laboratories. Various groups have demonstrated the role of autophagy in the regulation of normal SC functions. Autophagy plays a crucial role in the maintenance of pluripotency in embryonic SCs (ESCs) by maintaining pluripotency-associated proteins such as OCT4, SOX2, and Nanog. In 2014, Cho et al. demonstrated that autophagy is involved in the homeostasis of proteins that regulate pluripotency in hESCs [75]. Recent studies have suggested a prominent role for autophagy in the maintenance of quiescence and stemness of SCs and prevention of senescence by maintaining mitochondrial quality control [52,76]. Its role has been shown during neurogenesis through the management of oxidative stress responses and supply of metabolites to neural SCs [77,78]. In hematopoietic SCs (HSC), autophagy is required for the maintenance of stemness, which is mediated by a FOXO3A-induced survival program [79]. Autophagy also promotes the survival of mesenchymal SCs and human embryonic SCs [80,81]. Furthermore, autophagy is involved in the reprograming of somatic cells into induced pluripotent SCs, which is regulated by the pluripotency factor SOX2 that represses mTOR expression, resulting in increased autophagy [78,82]. Furthermore, mTOR inhibitors (i.e., rapamycin, pp242, or spermidine) that promote autophagy increase the efficiency of reprograming [83,84]. Interestingly, autophagy-mediated removal of mitochondria has been associated with the process of reprograming where inhibition of mTOR promotes mitophagy to limit mitochondrial content in an ATG5-dependent manner [85]. In contrast to these findings, Wu et al. demonstrated that mTORC1 impairs reprograming by triggering autophagy [86]. Their findings suggested that autophagy inhibits reprogramming by degrading p62, whose accumulation facilitates reprogramming in autophagy-deficient cells. Thus, a complex regulatory network exists between stem cell reprogramming factors, the mTOR pathway, and autophagy that regulates the reprogramming of somatic cells to induced pluripotent SCs.

### 3.4. Autophagy in the Regulation of Functions of CSCs

It is evident that autophagy, due to its role in quality control and cellular stress adaptation, plays an important role in regulating the properties of normal SCs. Similarly, the involvement of autophagy in the modulation of CSC properties and functions is also emerging [87]. In the following sections, we discuss the involvement of autophagy in various aspects of CSCs, which will not only help to understand tumor biology but also target such pathways by therapeutic interventions to improve clinical outcomes. CSCs have demonstrated higher autophagic fluxes in a variety of cancers than adherent cancer cells or normal cells [88,89,90]. Inhibition of autophagy via chemical modulators or siRNA-mediated inhibition of autophagy-regulating genes abolishes CSC properties such as stemness, self-renewal, and pluripotency propagation, further strengthening the role of autophagy in CSC maintenance. Similarly, inhibition of autophagy decreases the proportion of the side population, tumorsphere-forming ability, and expression of stemness genes [91].

Studies have indicated the role of Sonic Hedgehog (Shh) signaling and its contribution to CSC maintenance through the modulation of autophagy [92]. Observations by Wolf et al. identified an essential role of ATG4A in the maintenance of CSCs [93]. In line with these observations, Rothe et al. identified that several key autophagy genes were differentially expressed in CD34^+^ hematopoietic stem/progenitor cells, with a prominent role of ATG4B that was differentially expressed in pretreatment CML stem/progenitor cells from subsequent imatinib mesylate (IM) responders vs. IM non-responders (*p* < 0.05) and was helpful in predicting therapeutic response in CML [94]. Studies by Cufi et al. provided evidence for the role of autophagy in the maintenance of tumor cells expressing high levels of CD44 and low levels of CD24, which are features of BSCs [95]. Later, beclin1 was also identified to promote stemness and tumorigenesis in breast cancer [96]. The role of autophagy was further identified in the maintenance of BSCs by regulating IL6 secretion [97]. In line with this, it has been demonstrated that CSCs strictly require a basal level of autophagy to maintain a balance between pluripotency and differentiated/senescence stage [26]. Sharif et al. suggested a link between the nicotinamide adenine dinucleotide (NAD^+^) biosynthesis pathway, CSC transcription factor POU5F1, and pluripotency, where autophagy acts as a novel regulator of pluripotency of CSCs. Modulation of autophagy leads to loss of pluripotency and enhances the differentiation/senescence of CSCs [26]. Studies have demonstrated the role of the cancer/testis antigen DDX53 in promoting stem cell-like properties via autophagy, which further confers resistance to anticancer drugs in breast cancer cells [98].

Inhibition of Atg-5-mediated autophagy prevents cisplatin resistance by galectin-1 in hepatic cancer cells [99,100]. MiR-21 mimics in hepatic cancer cells restore sorafenib resistance by inhibiting autophagy [101]. BRAF increases the levels of autophagic markers, such as LC3 and BECN1, in colorectal cancer cells [102]. In an interesting study by Liu et al., the role of mitophagy controls the active levels of p53 to maintain hepatic cancer cell proliferation (details of this concept are discussed in the consecutive section) [9].

The study uncovered that increased mitophagy facilitates the degradation of p53. Conversely, when mitophagy is suppressed, PINK1 phosphorylates p53 at serine-392, leading to its relocation to the nucleus. In this location, p53 inhibits the transcription of Nanog, a protein crucial for maintaining the stemness and self-renewal capability of CSCs. CD133 (prominin-1), a marker for bioenergetic stress, is a well-identified marker of CSCs in a variety of solid tumors and has been identified to promote survival of CSCs by modulating autophagy [103]. Of note, it has also been demonstrated that CD133 promotes glucose uptake and autophagosome formation in addition to CSCs. It has also been shown that under conditions of glucose deprivation, CD133 increases the uptake of glucose and promotes the formation of autophagosomes.

## 4. Mitochondrial Autophagy: Mitophagy in CSCs

Mitophagy is a process of selective removal of old or dysfunctional mitochondria through the autophagic pathway and thus plays a crucial role in mitochondria quality, ROS production, and cellular metabolism. This process plays a critical role in regulating multiple-cellular signaling and processes such as cell growth, metabolic alterations, proliferation, quiescence, and cell death [104]. Due to the regulation of such crucial functions, mitophagy also plays a key role in the regulation of stem-cell functions.

Mitophagy is a multi-step process regulated by intrinsic and extrinsic signals involving a complex network of signaling molecules. The process of mitophagy includes the following: (1) process of initiation, (2) priming of mitochondria to be targeted for autophagy, (3) engulfment by autophagosomes, and (4) fusion with lysosomes followed by degradation by hydrolases [105]. The regulatory pathways of mitophagy are classified into PINK1/Parkin-mediated mitophagy and receptor-mediated mitophagy, which are initiated in response to different stimuli [105,106,107,108].

### 4.1. Mechanism of Mitophagy

One of the most well-characterized mitophagy pathways is represented by the phosphatase and tensin homolog (PTEN)-induced kinase 1 (PINK1)-PARKIN cascade. In damaged mitochondria that are depolarized, PINK1 accumulates and is stabilized, which is otherwise degraded by the matrix protein mitochondrial processing peptidase (MPP) and by the inner mitochondrial membrane (IMM) protein presenilin-associated rhomboid-like protease (PARL) under normal conditions. Stabilization of PINK1 results in phosphorylation of parkin, which stimulates PARKIN’s E3 ligase activity and recruitment to mitochondria. The PINK1 and parkin system now initiates ubiquitination of several OMM proteins that function as “eat-me” signals. These phosphorylated polyubiquitin chains on OMM proteins are recognized by various adaptor proteins (p62, optineurin, NDP52, NBR1, and TAX1BP1) that initiate binding with LC3 and mitophagosome formation. Furthermore, during mitochondrial damage, IMM proteins (cardiolipin (CL) and prohibitin 2 (PHB2)) are externalized to OMM and promote mitophagy. In addition, LC3 and other autophagosomal membrane proteins can interact with CL and other mitochondrial proteins such as BNIP3, under various conditions to initiate the process of mitophagy [105,107].

The role of mitochondrial fission and fusion in mitophagy regulation has also been demonstrated. This process involves the recruitment of dynamin-related protein 1 (Drp1) from the cytosol to mitochondria under stress conditions at the mitochondrial-associated endoplasmic reticulum (ER) membrane (MAM), followed by the inhibition of mitofusin 1 (Mfn1), mitofusin 2 (Mfn2), and optic atrophy 1 (OPA1), resulting in the dissociation of ER and mitochondria, thus promoting mitochondrial fragmentation and directing them toward degradation via mitophagy [105,107]. The downregulation of mitochondrial fusion protein 1/2 (MFN1/2) triggers Ras–Raf, and HIF1α signaling and promotes glycolytic metabolism that promotes cellular reprograming as well as pluripotency [109,110,111].

### 4.2. Mitophagy in Normal SCs

In iPSCs, the role of mitophagy has been associated with cell fate conversion, which involves a significant reduction in mitochondrial mass and numbers compared with parental somatic cells [112,113]. The crucial role of PINK1-dependent mitophagy has been identified in the reprograming of iPSCs and maintenance of pluripotency, indicating that mitophagy is directly responsible for determining the fate of SCs [114]. Similarly, inhibition of mitophagy via DRP1 reluctance suppresses the reprograming of mouse embryonic fibroblasts induced by Yamanaka factors (OCT4, SOX2, and KLF4) [77]. In hematopoietic SCs, quiescence and stemness are maintained by the removal of healthy mitochondria via mitophagy; thus, mitophagy preserves the regenerative capacity of old hematopoietic SCs [52]. Furthermore, structural and functional remodeling of mitochondria also regulates mitophagy and its associated cellular respiratory processes [115].

### 4.3. Mitophagy in CSCs

In cancer, the role of mitophagy is still obscure and has been demonstrated as a tumor promoter as well as a suppressor mechanism, depending on cancer type and stage. Its role has been associated with bioenergetics and metabolic switch that includes the shift toward the glycolytic phenotype in cancer cells and promotes metabolic plasticity [113,116,117]. In addition to the regulation of cancer cells, the role of mitophagy has been suggested in CSCs to regulate their stem cell potential and functions [9,11,118]. In an interesting study by Liu et al., the role of mitophagy was identified in promoting transcriptional activation of Nanog, a key transcription factor required for stem cell properties in hepatic CSCs [9]. They demonstrated that PINK directly phosphorylated p53, followed by autophagosome-mediated sequestration and degradation of p53. Because p53 downregulates Nanog, degradation of p53 results in the transcriptional upregulation of Nanog and related functions of stemness and self-renewal [9]. Reports have suggested that mitophagy plays a crucial role in the reduction of OXPHOS and in promoting the transition toward glycolysis, which acts as an important step in promoting the reprograming and stemness of SCs/CSCs [110,111,119]. Studies on pancreatic CSCs (PaCSCs) by Alcala et al. demonstrated the essential role of mitophagy in maintaining CSC functions [120]. They reported a role link between interferon-stimulated gene 15 (ISG15)/ISGylation, mitophagy, and metabolic plasticity that drive PaCSC functions essentially by maintaining their metabolic plasticity. ISG15 is involved in mitochondrial ISGylation and targets damaged and old mitochondria toward mitophagy. Loss of ISG15 not only results in the accumulation of damaged mitochondria via mitophagy and reduced oxidative phosphorylation (OXPHOS) along with glycolysis but also interrupts metabolic plasticity. Their study suggests targeting ISG15/ISGylation to prevent the metabolic plasticity of PaCSCs for preventing therapeutic resistance [120].

### 4.4. Mitophagy Encourages or Fosters the Stemness of CSCs

From past evidence, mitophagy has been shown to be required for the self-renewal of CSCs by turning over respiring mitochondria to maintain CSCs in a glycolytic state with low levels of oxidative metabolism [121,122]. The balance between glycolysis and oxidative metabolism has been reported in numerous systems to determine the rates of stem-cell quiescence versus differentiation [104,116,122]. In this regard, the upregulation of mitophagy-associated flux mediated by BNIP3L has been reported in human colorectal cancer-derived CSCs [123]. The role of mitophagy was also demonstrated in the self-renewal of human acute myeloid leukemia CSCs [118]. A higher rate of mitochondrial fission, an important step that promotes mitophagy, has also been associated with the promotion of the stemness of CSCs [16]. In hepatocellular carcinoma, mitophagy promoted the hepatic CSC population, whereas inhibition of mitophagy significantly inhibited such cells [9]. Furthermore, in liver CSCs, hepatitis B virus x protein (HBx)-induced BNIP3 L-dependent mitophagy resulted in metabolic reprogramming of HCC cells toward glycolysis and enhanced the stem-cell traits in HCC cells in vivo and in vitro [121]. In addition to promoting metabolic shift that sustains CSC survival and function under stress conditions, mitophagy flux confers chemoresistance. Upregulation of BNIP3L-mediated mitophagy has been identified as an escape mechanism against doxorubicin-induced cell death in CD133^+^/CD44^+^ CSCs derived from human colorectal cancer [123].

Therefore, it can be hypothesized that mitophagy provides resistance against therapeutic stress by clearing damaged mitochondria. This process could prevent the increase in reactive oxygen species (ROS), which might otherwise trigger programmed cell death. Consequently, mitophagy is proposed to play a balancing role in maintaining mitochondrial ROS levels and regulating the associated stress and cell-death pathways in CSCs.

## 5. Role of Autophagy in Regulating the Dormancy or Quiescence of CSCs

Tumor dormancy has been identified as one of the root causes of tumor relapse and has emerged as a potential therapeutic target to prevent metastasis and recurrence of the disease. Cancer cells adapt to the dormant stage to survive or resist adverse conditions during primary tumor formation, metastasis, and therapeutic stress as well as to escape from immune surveillance. Moreover, plasticity, therapeutic resistance, and dormancy have been proposed as interlinked processes that involve reversible genetic alterations [37]. Thus, dormancy of CSCs can be directly impacted by therapeutic interventions; however, it is not clear whether dormancy induction can be used as an approach to inhibit tumor metastasis or whether reactivation of the dormant cells to sensitize them toward therapy may be beneficial in eliminating such cells because the dormant CSCs can reenter the proliferating stage and may lead to tumor recurrence [38].

The relapse of the tumors even after two decades suggests the presence of a dormant population within the tumor cells that can proliferate when evoked [124]. It has been hypothesized that similar to normal tissue SCs, dormant tumor cells are basically tumor SCs or CSCs in a quiescent state, which can expand to re-populate the tumor [125]. In 2009, Dembinski and Krauss demonstrated a subpopulation of slow cycling cells in pancreatic adenocarcinoma cell lines that were also resistant to chemotherapy and exhibited increased tumorigenic and invasive potentials. Similarly, quiescent CSCs were also isolated from breast cancer, ovarian cancer patient specimens, liver cancer, and a human colon adenocarcinoma cell line [59,126,127].

### 5.1. Signaling Pathways Regulating Tumor Dormancy

Cellular quiescence is a reversible process in which cells enter and rest in the G0 phase and can re-enter the cell cycle after receiving a proliferation signal [127,128]. These cells are slow dividing, maintain low ROS levels, show chemoresistance, and are highly tumorigenic. The quiescent cells adapt to various long-term survival strategies for their maintenance and properties, such as retention of genomic integrity and ability to regain proliferation. Multiple signaling pathways regulate and maintain quiescent CSCs and their re-entry in the proliferating state [129,130]. These signaling molecules include the tumor suppressors p53 and retinoblastoma protein (RB), cyclin-dependent protein kinase inhibitors, namely, p21, p27 and p57, Notch-related pathways, and some micro-RNAs [131,132,133]. Several transcription factors, including forkhead box O (FOXOs) and nuclear factor 1 (NF1) protein member NFIX, have also been involved in gene expression regulation in quiescent cells [129].

### 5.2. Dormant State in CSCs

The presence of CSCs in the dormant or quiescent state has been one of the biggest hindrances to the action of anti-proliferating agents during therapy. This leads to CSC-related resistance and escape from radiotherapy and chemotherapeutic drugs. Moreover, these cells can also resist harsh tumor microenvironments and regain proliferation upon receiving the appropriate signal, leading to tumor regrowth and relapse. Hence, inducing the re-entry of these quiescent cells into the proliferating state can allow most of the anticancer agents that are antiproliferative (mitotic inhibition, antimetabolite drugs, and topoisomerase inhibitors) to act on these cells (Figure 5). In this regard, a previous study demonstrated that treatment with mitogens (GCSF) can sensitize leukemia CSCs to cell cycle-dependent chemotherapy [129,134]. In addition, ablation of the F-box protein Fbxw7, which inhibits ubiquitin-dependent degradation of c-Myc, Notch, and cyclin E, and thus promotes reentry of quiescent CSCs in the cell cycle, increases the sensitivity of Phi^+^ leukemia CSCs toward imatinib [135,136]. However, to date, this approach is not well established and has not been validated. Alternative approaches include the inhibition of proliferating CSCs to enter the quiescent state or the inhibition of the re-entry of quiescent CSCs into the proliferating stage after treatment to prevent tumor regrowth by inducing differentiation, senescence, or cell death in these cells (Figure 5). Thus, it is imperative to understand the pathways involved in the regulation of quiescent CSCs.

### 5.3. Regulation of CSC Dormancy by Autophagy

The role of autophagy in tumor dormancy has long been suggested to promote survival by removing damaged organelles and nutrient recycling of amino acids and nucleotides in a variety of cancers [54,111,137,138,139]. In dormant cells, autophagy also promotes chemoresistance, as demonstrated in a variety of cancers, including colorectal cancer, liver cancer, brain tumors, and melanoma [38,136,140]. Autophagy can play a crucial role in quiescent CSCs by providing basic requirements for survival, such as amino acid and ATP production, and preventing energetic catastrophe [137]. Data from various preclinical studies confirmed the link between autophagy and quiescence of cancer cells, where autophagy promotes survival in quiescent cells [38,83]. In accordance, in vivo findings also suggested that suppression of autophagy in quiescent CSCs decreases tumor regrowth and metastatic burden. Furthermore, inhibition of autophagy results in the accumulation of damaged organelles and ROS, resulting in apoptotic cell death of quiescent CSCs [127,140]. In addition, studies from ovarian cancer patients undergoing clinical remission after surgery and chemotherapy demonstrated the upregulation of autophagy in the dormant cancer cells in their tissue specimens [141]. In addition, transcriptional upregulation of autophagy-related genes (ATGs), such as LC3, ATG4, ATG5, ATG7, and BECLIN1, has been shown in quiescent CSCs. The involvement of AMPK signaling and mTOR in regulating autophagic pathways in such cells has also been identified [101,102]. Wang et al. demonstrated the role of autophagy in promoting self-renewal and maintenance of quiescence in ovarian cancer spheroid cells with CSC properties [59]. Their study identified that a positive loop is formed between autophagy and the NRF2 signaling pathway, which promotes the antioxidant response in ovarian cancer spheroid cells, and inhibition of the autophagic pathway disturbs the antioxidant response and self-renewal capacity of the spheroid cells and their tumorigenesis capacity [59]. In ovarian cancer cells, the role of the ARHI protein (aplasia Ras homolog member I), which is a tumor suppressor, has been identified in the induction of cancer-cell dormancy through the induction of autophagy by inhibiting the phosphatidyl inositol 3-kinase (PI3K)-protein kinase B (AKT) growth-factor signaling pathway [101]. Similarly, autophagy inhibition via BECLIN1, ATG5, or chloroquine treatment on GSCs enhanced the expression of stemness markers (e.g., CD133, POU5F1, SOX2, BMI1, LGR5, and Nanog), along with increased proliferation and clonogenicity. This suggests a departure from a dormant stemness status to an active, proliferating state when autophagy is inhibited [132]. This study suggested that autophagy played a major role in tumor dormancy in such tumors and that a coordination between autophagy and growth arrest exists that promotes cell survival and possibly maintains the stem-like state of quiescent SC/CSCs.

### 5.4. Autophagy in the Regulation of Therapy-Induced Dormancy of CSCs

It has been observed that various chemotherapeutic agents also trigger to enter the quiescent state. In the case of gastrointestinal stromal tumors (GISTs), treatment with IM, a tyrosine kinase inhibitor, triggers quiescence via activation of autophagy, which is mediated by assembly of the DREAM complex and inhibition of the PI3K/AKT pathway [137,138]. However, these quiescent cells could reenter the cell cycle and expand once imatinib was removed. Furthermore, in cases of chronic myeloid leukemia (CML), leukemic SCs (LSCs) show resistance to tyrosine kinase inhibitor treatment via induction of the autophagy-mediated survival pathway [142]. Furthermore, Baquero et al. identified that long-term hematopoietic SCs (HSCs) isolated from leukemic mice maintain higher basal levels of autophagy than non-leukemic long-term subsets of HSCs and more mature leukemic cells. They also demonstrated that Lys05, a highly potent lysosomotropic agent, mediated autophagy inhibition and reduced LSC quiescence followed by the expansion of myeloid cells. Furthermore, Lys05 and PIK-III, when used in combination with tyrosine kinase inhibitor treatment, were effective in targeting primary CML LSCs and xenografted LSCs.

Given the role of autophagy in CSCs and in promoting tumor cell survival, several groups have proposed that dormant tumor cells depend on autophagy to survive at secondary sites over extended periods of time and grow out later as macrometastases [129,130]. Thus, autophagy is required to sustain the dormant or quiescent state of CSCs and to maintain their stemness and appears as an interesting target to modulate therapeutic resistance of these cells. The timing of administration of autophagy inhibitors is critical because if these inhibitors are administered before chemotherapy, they might impair the formation of quiescent cancer cells and decrease autophagy [124].

Overall, it is evident that autophagy is activated to support the metabolic functions of dormant tumor cells and not only promotes survival but also contributes to therapy resistance. It is important to consider these observations, which provide a strong basis for combining genotoxic therapy with autophagy inhibition that would preferentially eliminate dormant tumor cells and associated metastatic dormancy and tumor relapse.

## 6. Autophagy in the Metabolic Regulation of CSCs

To meet the bioenergetic, biosynthetic, and redox demands of cancer cells, various genotypic or phenotypic changes are both direct and indirect consequences of oncogenic mutations. Cancer cells display distinct metabolic phenotypes compared with their normal counterparts, which are referred to as “metabolic reprogramming” and are a “hallmark of cancer cells” [143]. Interestingly, heterogeneity among the tumor cell types was also observed at the metabolic level.

### 6.1. Metabolic Reprogramming of Cancer Cells

Because of metabolic reprogramming, changes in both intracellular and extracellular metabolites are observed in cancer cells that further regulate gene expression, cellular differentiation, and the tumor microenvironment. By undergoing metabolic changes, cancer cells develop the advantage of acquiring necessary nutrients from a nutrient-deficient environment to maintain their viability and proliferative state. Metabolic reprogramming of cancer cells mainly includes deregulated uptake of glucose and amino acids, nutrient acquisition, use of glycolysis/TCA cycle intermediates for biosynthesis and NADPH production, increased demand for nitrogen, altered gene regulation, and interaction with the microenvironment. Such changes promote tumorigenesis by facilitating and enabling processes required for rapid proliferation, survival, invasion, metastasis, and resistance to therapies [143].

### 6.2. Metabolic Alteration in CSCs

Notably, cancer stem cells (CSCs) possess a unique metabolism that differs from that of non-CSCs, enabling them to sustain their stem-like characteristics. In CSCs, these metabolic pathways are regulated by various signaling cascade, including Hippo, WNT/β-catenin, JAK/STAT, and Notch [144]. Contradictory reports complicate matters by indicating that CSCs might favor glycolysis or depend on oxidative phosphorylation (OXPHOS) for their energy needs. It is evident that CSCs display metabolic flexibility, relying on glycolytic and/or oxidative metabolism depending on the microenvironment and energy demands [145].

In normoxic tumors, heterogeneity is observed in terms of metabolic pathways adapted by CSCs; upregulation of glycolytic enzymes and dependence on mitochondrial pathways as well as mitochondrial fatty acid oxidation (FAO) for generation of ATP and NAD^+^ has been observed [146]. However, under hypoxia, glycolysis is upregulated and is mediated by HIF-1α that promotes upregulation and activation of several glycolytic proteins, including glycolytic enzymes and glucose transporters [147]. Interestingly, CSCs induced by epithelial-to-mesenchymal transition demonstrate higher uptake of extracellular catabolites, such as pyruvate, lactate, glutamine, glutamate, alanine, and ketone bodies [148]. Furthermore, quiescent disseminated tumor cells rely on alternative energy sources such as autophagy [128].

### 6.3. Role of Autophagy in Regulating the Metabolic Pathway of CSCs

The role of autophagy in regulating cellular metabolism under physiological and pathological conditions is well established [15,54,128,149,150]. Autophagy is often used as an alternative pathway to meet the metabolic demands of cells under conditions of stress or nutritional and oxygen deprivation. Tumor cells activate autophagy as a cellular stress response or to meet increased metabolic demands. The role of autophagy in promoting cell survival has also been attributed to its role in energy production, which promotes tumor growth and therapeutic resistance [151]. Interestingly, earlier studies have demonstrated that autophagy and microlipophagy are as critical as electron transport chain activity and represent a strategy for SCs to maintain their energetic balance, which is crucial for their survival. Similar to SCs, CSCs are also metabolically distinct from their differentiated counterparts [152]. Various studies have provided evidence that autophagy governs metabolic pathways in CSCs [153]. CD133-expressing GSCs show better survival under nutrient deprivation conditions. It was suggested that CD133 is involved in the autophagic process under such conditions, translocates to the cytoplasm, contributes to the membrane source of the autophagosomes, is ultimately degraded by lysosomes, and promotes cell survival [111].

### 6.4. Autophagy and Metabolic Plasticity of CSCs

Emerging data suggest a role of autophagy as a key regulator of metabolic plasticity that promotes the metabolic adaptability of CSCs in the vigorous microenvironment and enables these cells to survive in hypoxic, nutrient-deficient niches [13]. Autophagy regulates CSC metabolism by controlling the cellular redox state, lipid metabolism, and dependency of CSCs on amino acids or ketone bodies and other metabolites [13,153,154]. It has been shown that high-energy metabolites such as lactate and ketones promote tumor growth and metastasis [155] and promote CSC stemness [156]. The role of autophagy has been shown to be involved in lactate production and secretion in heat stress-immature Sertoli cells [157]. Inhibition of autophagy decreased the proportion of CSCs and glycolytic gene expression in urothelial carcinoma (UC) cells, suggesting that autophagy could provide energy and nutrients for CSCs to maintain their stemness [158].

The metabolic plasticity of CSCs allows them to produce energy through various pathways that not only promote survival and support metastatic growth but also provide resistance under various adverse conditions and therapeutic agents [144]. In an important study, based on transcriptomic and metabolomics analysis data, a molecular link between autophagy and metabolic mechanisms was suggested in CSCs, where autophagy supports the survival of these cells [87]. This study illustrated that in pancreatic ductal adenocarcinoma (PDAC), even after oncogene ablation of mutated KRAS and p53, the primary drivers of PDAC, a subset of quiescent tumor cells exhibiting characteristics of CSCs persists. These cells are accountable for tumor recurrence. Based on transcriptomic and metabolic analyses of these cells, the expressions of various genes involved in mitochondrial function, autophagy, and lysosomal activity were identified. Importantly, strong reliance on mitochondrial respiration and decreased dependence on glycolysis for cellular energetics were identified as prominent features of these surviving CSCs. Targeting mitochondrial respiration significantly inhibited the survival of these cells and hampered their tumorigenic potential. Overall, the study suggested that mitochondrial electron transport activity was strongly dependent on autophagic processes. Furthermore, in another study, it was observed that deletion of an SRC activator and neural precursor cell expressed developmentally downregulated 9 (NEDD9), a scaffolding protein that is crucial for tumorigenesis and metastases, in KRAS, and Trp 53 mutated NLCSCs showed elevated levels of LKB1 and AMPK, which fuel tumor growth by increasing autophagy [159]. Therefore, targeting the KRAS pathway along with mitochondrial respiration appears to be a potential target for the elimination of bulk tumor cells along with the dormant CSC population [87].

### 6.5. Autophagy Regulates Metabolic Adaptations in the Tumor Microenvironment and Cancer Stem Cells

In addition to the direct regulation of CSC metabolic functions, metabolic reprograming is also observed in cancer-associated fibroblasts, which are key components of the CSC microenvironment that rely more on aerobic glycolysis than oxidative phosphorylation [120]. Simultaneously, these cells also show upregulation of autophagic programs to support their proliferative and migratory capability, along with secretion of cytokines and growth factors [160,161]. Moreover, autophagy-derived catabolic substrates from cancer-associated fibroblasts also support the energy needs of pancreatic ductal adenocarcinoma [150,162]. An important role of autophagy has been observed during nutrient-poor states during which quiescent disseminated tumor cells rely mainly on autophagy to meet the energy demand and promote cell survival in harsh environments.

Overall, autophagy upregulation in CSCs promotes metabolic homeostasis and survival under various harmful conditions faced by these cells, such as starvation, energy deficiency, hypoxia, or anticancer treatment.

## 7. Crosstalk between Autophagy, CSCs, and the Tumor Microenvironment

CSCs reside and thrive in established niches that are composed of well-organized cellular and acellular milieus. These niches comprise various stromal and immune cells that lead to effective immune evasion, extracellular matrix remodeling, and angiogenesis [163]. Multiple components of the microenvironment, such as hypoxia, nutrient availability, oxidative stress, pH, and immune cells, govern the functions of CSCs, such as self-renewal and plasticity. In fact, CSCs lose their capacity for continued self-renewal when removed from their niche, suggesting the crucial role of the tumor microenvironment in the regulation of CSCs [164]. The role of autophagy in regulating the tumor microenvironment has been identified; however, the crosstalk between CSCs, non-CSCs, and autophagy regulation needs more clarity [45]. Several researchers have demonstrated that tumor cells induce autophagy in the microenvironment, which results in the availability of recycled nutrients for tumor growth. It has been suggested that autophagy plays an active role in regulating metabolic crosstalk between CSCs and non-CSCs within the tumor niche, resulting in metabolic symbiosis [165]. Therefore, it can be postulated that targeting CSCs and non-CSCs with autophagic inhibitors may disrupt metabolic symbiosis, leading to a negative influence on CSC properties. Indeed, inhibition of autophagy through oral administration of chloroquine is more effective than its localized inhibition within the tumor [65,166]. Another crucial component of the microenvironment is hypoxia, a well-known regulator of autophagy mediated by hypoxia-inducible factor 1 alpha (HIF-1α). Indeed, hypoxia-induced autophagy is crucial not only for the survival of liver CD133+ CSCs but also for the balance between non-stem pancreatic cancer cells and pancreatic CSCs [167]. In multiple human AML cell lines and primary blasts, hypoxia results in the upregulation of autophagy, and the inhibition of autophagy can sensitize otherwise resistant leukemia SCs (LSCs) toward chemotherapy [168,169]. Furthermore, the role of autophagy has also been demonstrated in the evasion of immune response, leading to immunosuppressive-related chemoresistance [170,171].

## 8. Crucial Role of Autophagy in the Therapeutic Resistance of CSCs

With advancements in the understanding of CSC biology, it is evident that these cells are responsible for tumor recurrence as they acquire resistance to therapy and the ability to remain or enter the quiescent stage. Drug-resistance mechanisms that are active in these cells include the following: (1) highly efficient and active DNA damage repair mechanism; (2) higher levels of MDR gene expression leading to drug efflux; (3) cellular plasticity; (4) higher levels of aldehyde dehydrogenase; (5) efficient antioxidant defense mechanism; (6) escape from apoptosis; (7) immunomodulation to escape immune cells; (8) microenvironment of CSCs; and (9) autophagy regulation. The role of autophagy is emerging because of its regulatory role in the therapeutic resistance of CSCs.

### 8.1. Chemoresistance: Autophagy Promotes Treatment Resistance

Autophagy plays a proven role in promoting survival under stressed conditions and provides resistance to chemotherapeutic stress. Our group has extensively studied the role of autophagy in promoting cell survival under pathological stress conditions [172,173,174,175]. Upregulation of autophagy during therapeutic stress is associated with the removal of damaged organelles and molecules and recycling of cellular contents, thereby promoting survival. Upregulation of autophagy has also been associated with the chemo-resistant properties of CSCs [13,131].

Ovarian CSC CD44/CD117^+^ cells demonstrated higher basal levels of autophagy than their non-stem counterparts. Genetic or chemical inhibition of autophagy impaired canonical CSC properties, such as viability, the ability to form spheroidal structures in vitro, and in vivo tumorigenic potential. Furthermore, inhibition of autophagy demonstrated a synergistic effect with carboplatin administration on both in vitro CSC properties and in vivo tumorigenic activity [176]. A recent study by Zhu et al. demonstrated a novel mechanism by which the SOX2/β-catenin/Beclin1/autophagy signaling axis regulates chemoresistance, stemness, and EMT in colorectal cancer (CRC) [177]. It was demonstrated that SOX2 along with β-catenin, regulates transcriptional activation of Beclin1, which leads to the activation of autophagy, followed by ABCC2 expression. These findings were also validated in CRC patients as well as xenograft models, which confirmed that inhibition of SOX2 expression and autophagy restrained tumor growth and chemoresistance in vivo [177]. Together, this study demonstrated a crucial role of SOX2, a potential driver of SCs in regulation of autophagy, leading to chemoresistance. Our research group has also shown that autophagy activation in cancer stem cells (CSCs) confers resistance against temozolomide treatment [172]. Even more intriguingly, the transformation of non-CSCs into CSCs induced by temozolomide (TMZ) was found to be facilitated by autophagy. This implies a significant involvement of autophagy in promoting CSC plasticity [172].

In 2009, while extensive screening of compounds preferentially toxic against CSCs, Gupta et al. identified salinomycin, a broad-spectrum antibiotic, as the most efficient agent, having more than 100-fold efficacy compared with paclitaxel, to kill BSCs in mice [140]. Interestingly, salinomycin induces the death of CSCs by suppressing autophagic flux [178,179]. In GSCs, a combination of bevacizumab, an EGFR inhibitor, or TMZ along with chloroquine, a late-stage autophagy inhibitor, has demonstrated improved drug sensitivity in the GSC population [180,181]. Similarly, inhibition of autophagy with chloroquine increased the susceptibility of pancreatic CSCs to gemcitabine [182]. Inhibition of autophagy also sensitized the CSC population in triple-negative breast cancer (TNBC) cells toward paclitaxel, which involved inhibition of the Janus-activated kinase 2 (Jak2)-signal transducer and activator of transcription 3 signaling pathway [183]. In gastric CSCs, combining 5-fluorouracil, chloroquine, and Notch inhibitors significantly improves the treatment efficacy associated with reduced CSC viability [184]. Furthermore, in head and neck squamous cell carcinoma (HNSCC), afatinib, a second-generation tyrosine kinase inhibitor, induced mTOR complex 1 (mTORC1) suppression-mediated autophagy. Pharmacological or genetic inhibition of autophagy sensitized HNSCC cells to imatinib-induced apoptosis, demonstrating that afatinib activates pro-survival autophagy in HNSCC cells [185]. Recently, Shi et al. demonstrated that nicardipine, a dihydropyridine calcium channel antagonist, can sensitize GSCs toward TMZ by inhibiting autophagy via mTOR modulation [186]. Further, a study by Wei et al. identified that photodynamic therapy (PDT), which has been associated with the recurrence and progression of colorectal cancer, leads to the upregulation of protective autophagy in CSCs, and the inhibition of autophagy enhances the PDT sensitivity of CSCs [187].

Overall, the role of autophagy enhances resistance to conventional cancer treatment in CSCs. This indicates autophagy inhibition as a potential target for combinational therapies to improve the efficacy of conventional cancer treatment modalities.

### 8.2. Inhibitory Role of Autophagy in CSCs: An Alternate View

In addition to its role in promoting cell survival, autophagy has also been reported to play a crucial role in cell-death pathways. Its role has been reported in tumor suppression by removing transformed cells with deleterious mutations as well as damaged cellular content by the process of autophagosome formation followed by lysosome-mediated degradation. Autophagic cell death is defined as programmed cell death-II; however, the criteria for differentiating autophagic cell death remain controversial. It has been suggested that uncontrolled overactivation of autophagy and/or selective removal of autophagy substrates are associated with such cell-death pathways [188]. Interestingly, in addition to its role in promoting CSC survival and maintenance, upregulation of autophagy has been shown to inhibit CSC properties by promoting differentiation, senescence, or cell death, thereby promoting drug-mediated cytotoxicity [10]. Zhuang et al. reported that curcumin, a natural compound with anti-inflammatory potential, induces lethal autophagy and exhibits differentiating and tumor suppressing properties in GSCs, thus inhibiting tumor formation capacity in vitro and in vivo [189]. Similarly, studies by Zhuang et al. (2011) demonstrated that autophagy induction by rapamycin triggers the differentiation cascade and promotes radiosensitization of GSCs [190]. The tumor-suppressive property of rapamycin was also reported in the murine S180 sarcoma model by suppression of the CSC phenotype via upregulation of autophagy [191]. Other agents, such as cannabidiol, trigger GSC differentiation by activating the autophagic process and inhibit proliferation and clonogenic capability by activating the transient receptor potential vanilloid-2. Their combination abrogates BCNU chemoresistance in GSCs [192]. Studies have further identified that histone deacetylase inhibitors induce autophagy, which results in differentiation and autophagy-mediated cell death in GSCs [193]. Inactivation of mTOR, a well-known inducer of autophagy, stimulates the differentiation of neuroblastoma and GSCs [62,194]. Tao et al. demonstrated that upregulation of autophagy by inhibition of mTOR suppressed self-renewal ability and tumorigenicity in CD133+ and Nestin+ GSCs, whereas inhibition of autophagy at an early stage by the PI3K inhibitor 3-MA reversed the mTOR inhibition phenotype [195]. Corroborating findings were observed using mTOR suppression treatment in mice, which resulted in tumor-size reduction and prolonged survival through degradation of Notch1, which is associated with maintenance of the self-renewal ability of GSCs [195]. Furthermore, resveratrol inhibits BSC survival by inhibiting the Wnt pathway, which leads to autophagy induction [196]. The role of autophagy in inhibiting dedifferentiation in a hepatocyte model to study the cell of origin of hepatocellular carcinoma (HCC) has also been identified. Following chronic liver injury, hepatocytes dedifferentiate into liver progenitor cells, which can be repressed by autophagy inhibition [197]. All these findings support the alternate idea that autophagy inhibits CSC stemness and proliferation.

Segala et al. demonstrated novel pharmacological control of lethal autophagy by a nuclear receptor through the action of a cholesterol metabolite [198]. They identified dendrogenin-A, a cholesterol metabolite that inhibits 3β-hydroxysterol-Δ8,7-isomerase and results in Δ8-sterols accumulation, can also act as an agonist of the Liver-X receptor and lead to the expression of Nur77, Nor1, and LC3, the pro-autophagic proteins. Thus, dendrogenin-A can activate multiple autophagic regulatory circuits, which can lead to lethal autophagy in cases of melanoma and acute myeloid leukemia both in non-CSC and progenitor cells. In another study, in the case of lung CSCs, inhibition of Stearoyl-CoA-desaturase 1 activity reverts resistance to cisplatin in lung CSCs by upregulating autophagy [199]. A recent study also demonstrated that posaconazole, which has demonstrated antitumor activity for glioblastoma, inhibits the stemness of cancer stem-like cells by inducing autophagy and suppressing the Wnt/β-catenin/survivin signaling pathway in both in vitro and in vivo models of glioblastoma [10]. Using autophagy inhibitors, chloroquine or Atg5 shRNA, the suppressive effect of posaconazole on CSC stemness was partially relieved. Moreover, as mentioned earlier, inhibition of autophagy downregulates stemness and promotes the differentiation of quiescent CSCs.

Taken together, it is essential to consider several factors such as timing of treatment, combinational agents, type, and stage of cancer before targeting autophagic pathways in CSCs as a part of the treatment regime [200]. Moreover, the clinical implications of inhibiting autophagy require careful consideration of CSC heterogeneity, acute adaptability, and plasticity [201]. Given the diverse roles of autophagy, any strategies aimed at targeting it must be thoughtfully planned, taking into consideration factors such as optimal timing, dosage, and potential drug combinations.

### 8.3. Role of Autophagy in CSCs in Response to Radiotherapy

Radiation therapy or radiotherapy is used for cancer treatment, in which high doses of radiation are used to kill tumor cells. Radiotherapy targets tumor cells with photons (e.g., X-ray), and particles (e.g., electrons, protons, and heavy ions), which damage DNA either directly, or lead to irreparable DNA damage, or by ROS generation via its reaction with water molecules in the cells. Approximately 50% of patients receive radiation therapy alone or in combination to kill or shrink cancer cells [202]. However, radioresistance remains a major obstacle that triggers the transformation of non-cancerous cells to cancerous genotypes. Non-CSCs are radiosensitive and are eliminated by radiation therapy; however, the smaller population of radioresistant CSCs in heterogeneous tumors poses a major obstacle to radiation therapy [152,203]. To add further complexity, in response to radiation therapy, the non-CSC population can dedifferentiate and convert to the CSC population, making the treatment even more challenging [204,205]. Thus, radiation therapy may further contribute to the generation of a radioresistant population of CSCs that may contribute to therapy resistance, relapse, and metastasis of the tumor. Thus, identification of (1) the basis of CSC radioresistance, (2) transformation of non-CSCs to CSCs, and (3) enhancement of the sensitivity of heterogeneous tumor populations toward radiotherapy is the major focus of various investigators.

It has been identified that CSCs are radioresistant, which is associated with their DNA damage-repair capacity, antioxidant mechanism, quiescence (enter into non-diving stage), anti-apoptotic machinery, and autophagy [152,206]. In 2006, Bao et al. identified that CD133-positive tumor cells in gliomas were enriched after radiotherapy and were responsible for radioresistance. This radioresistance was attributed to the DNA repair mechanism in CSCs. They further identified that targeting DNA repair checkpoints can sensitize CSCs to radiotherapy [203]. Autophagy upregulation has been reported with radiation treatment in a variety of CSCs in brain, breast, prostate, and liver cancers; however, its role in radioresistance or radiation-induced cell death is debatable [152]. Following radiation therapy, autophagy is induced in BSCs and is associated with radioresistance and enrichment of BSC subpopulations [91,207].

Various pathways induced by radiation therapy, including PI3K/Akt/mTOR, DNA-PK, tumor suppressor genes, mitochondrial damage and lysosomes, may play an important role in radiation-induced autophagy in glioma cells [206]. Gamma radiation induced a significantly higher level of autophagy in GSCs than in non-GSCs. Further inhibition of autophagy by chemical agents or siRNA improved the radiosensitivity of GSCs [208]. In support of the role of autophagy in radioresistance, it was identified by Chang et al. (2014) that radioresistant prostate cancer cells were arrested in the G0/G1 and S phases of the cell cycle along with activation of the cell-cycle checkpoint, autophagy, and DNA repair-pathway proteins, along with inactivation of the cell-death pathway [152]. Interestingly, this group studied the combination of dual PI3K/Akt/mTOR inhibitors (BEZ235 or PI103) with radiotherapy treatment in CSCs [209]. They identified that this dual inhibitory effect greatly improved the treatment efficacy by repressing colony formation, enhancing apoptosis, leading to the arrest of the G2/M phase, increased double-strand break levels, and less inactivation of cell-cycle checkpoint, autophagy, and non-homologous end joining (NHEJ)/homologous recombination (HR) repair-pathway proteins in CaP-radioresistant cells. Huang et al. suggested the role of the MST4-ATG4B signaling axis in influencing GBM autophagy and malignancy [181]. They identified that MST4 regulates ERK signaling and activation of ATG4, which in turn activates ATG8, leading to autophagosome formation. Inhibition of this signaling axis inhibits GSC tumor formation and confers radioresistance in GSC. A later study by the same group demonstrated a role for MIR93 as a regulator of autophagy in GSCs [210]. MIR93 expression significantly reduces the proliferation, sphere-forming frequency, and sphere-formation ability in GSCs and sensitizes them toward TMZ or IR by upregulating apoptotic cell death [210]. Recent studies further confirmed the activation of autophagy along with unfolded protein response in a radioresistant population of GSCs with an involvement of overactivation of ER stress-related pathways. Using the ER stress-inducing drug 2-deoxy-D-glucose (2-DG) improved the sensitivity of these resistant populations of GSCs toward radiation with an increase in apoptotic cell death [211].

Zhuang et al. reported that the induction of autophagy in patient-derived GSCs leads to the differentiation of CSCs that are further sensitized to radiation therapy both in vitro and in vivo [190]. A combined effect of an autophagy inducer (rapamycin) and radiation treatment significantly reduced tumor-forming ability, tumor growth, and associated mortality in mice after intracerebral grafting of human GSC/GIC [190]. Similarly, the integrin inhibitor cilengitide decreased cell viability and self-renewal of GSCs by inducing autophagy and sensitized them towards toward-irradiation, suggesting a role for autophagy induction in cilengitide-induced cytotoxic effects and radiosensitization [212]. In renal cell carcinoma (RCC), STF-62247, an agent causing autophagic cell death, when combined with radiation increases cell death under oxic and hypoxic/physiological conditions [213].

A recent study by Yang et al. demonstrated that FLASH-IR (∼109 Gy/s), at a dose of 6–9 Gy via laser-accelerated nanosecond particles induced apoptosis, proptosis, and necrosis in both CSCs and normal cancer cells; however, CSCs were more resistant to radiation than normal cancer cells [60]. The radioresistance of CSCs was mainly associated with an increase in lysosome-mediated autophagy and a decrease in apoptosis, necrosis, and pyroptosis. Interestingly, another study demonstrated that fractionated IR could induce the stemness phenotype of GSCs, which is mediated by a P62-dependent pathway involving activation of the Wnt/β-catenin pathway and such a population of GSCs promotes radioresistance and is associated with poor prognosis [214]. Moreover, this study shows that ATG5-mediated autophagy is activated in response to glutamine depletion as a tumor-survival strategy to withstand radiation-mediated cellular damage.

Similarly, glutaminase-driven glutamine catabolism has also been shown to regulate prostate cancer radiosensitivity by regulating the redox state, stemness, and ATG5-mediated autophagy, where autophagy inhibition promotes radiosensitivity of prostate CSCs [215]. In ovarian CSCs subjected to ionizing radiation at a cumulative dose of 8 Gy, protective autophagy was upregulated. Treatment of these cells with lumiflavin enhanced the effects of ionizing radiation on ovarian CSCs by blocking the autophagy pathway [216].

Furthermore, the role of autophagy has been identified in acquired radioresistance (ARR), which is one of the important reasons for the failure of radiotherapy. Long-term fractionated irradiation (FI) treatment leads to an acquired radioresistance phenotype in the bladder cancer cell model, where autophagy plays a critical role [82]. Inhibiting autophagy could sensitize these radioresistant cells toward taxol treatment, further validating the role of autophagy in promoting chemical tolerance in these resistant CSCs [82].

## 9. Conclusions

It is evident that although autophagy plays an essential role in the regulation of CSCs, its role is complex and can vary depending on the cell type and multiple factors. A clear understanding of the complex regulatory network will help harness the potential of autophagy for developing clinically effective antitumor strategies. The role of autophagy has emerged in not only self-renewal but also promoting the survival of CSCs under adverse conditions. As shown in Figure 6, autophagy also regulates the quiescent state, metabolic plasticity, and other therapy resistance pathways in CSCs. Autophagy is also involved in the reprograming of non-CSCs to undergo dedifferentiation to CSCs in response to various therapeutic agents; inhibition of such conversions can act as a potential target. Furthermore, because of the emerging role of autophagy in regulating the crosstalk between CSCs and their microenvironment, it is imperative to understand its role in regulating the stromal cells, endothelial cells, and tumor-infiltrating innate and adaptive immune cells that ultimately regulate CSCs and their plasticity. Identification of inhibitors/activators of autophagy that can target specific autophagy pathways without pleiotropic effects may be more useful. Moreover, because autophagy is a physiological process that plays an essential role in maintaining the healthy state of a cell, it becomes important to consider that therapeutic interventions targeting autophagy can be associated with multiple side effects.

## Figures and Tables

**Figure 1 cells-13-00447-f001:**
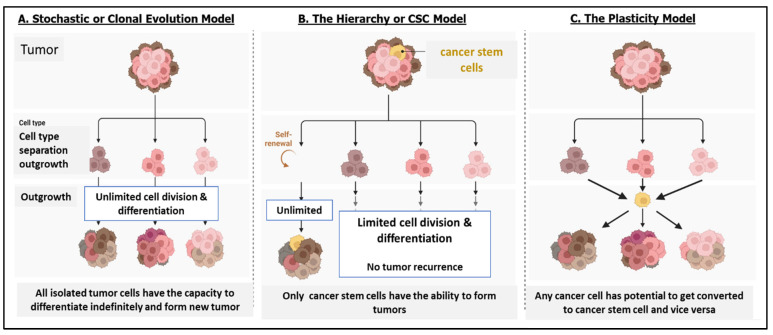
Diagrammatic representation of various models leading to carcinogenesis. (**A**). The stochastic or clonal evolution model, which suggests that normal cells undergo a series of mutations and transform into cancer cells that possess the capacity to form a bulk tumor. (**B**). The hierarchy model suggests that tumors originate from CSCs that are pluripotent and self-renewing. (**C**). The plasticity model suggests that differentiated cells can dedifferentiate into CSCs and that there is a dynamic interconversion between CSCs and non-CSCs, leading to tumor heterogeneity. (The template for figures was adapted from Biorender.com).

**Figure 2 cells-13-00447-f002:**
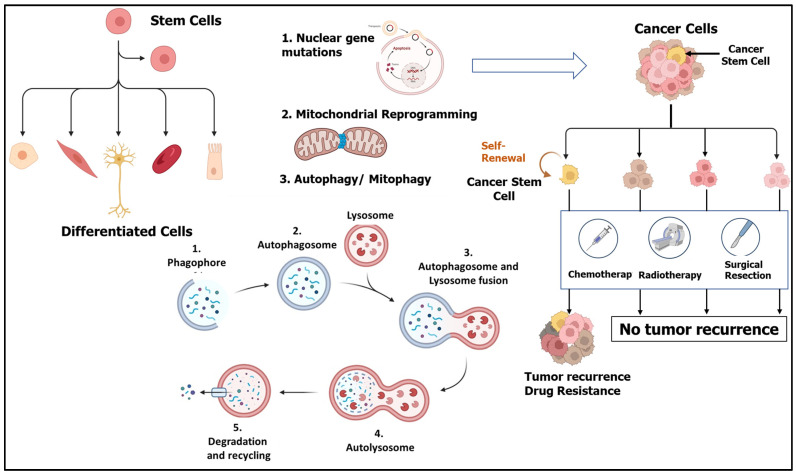
An overview of cancer progression. Stem cells or differentiated cells may undergo various types of mutations, including genetic mutations such as the p53 mutation, mitochondrial reprogramming, and/or autophagy/mitophagy alterations, to transform into cancer cells. Among these heterogeneous cancer subpopulations, CSCs are considered to be the main cause of tumor recurrence and drug resistance. (The template for figures was adapted from Biorender.com).

**Figure 3 cells-13-00447-f003:**
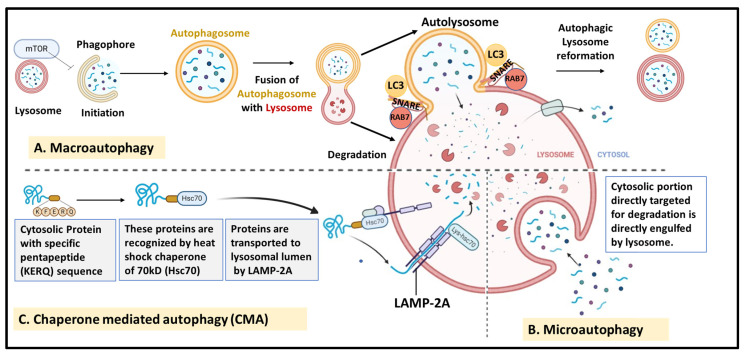
Types of autophagy based on substrates and regulatory proteins. (**A**). Macroautophagy, in which cellular components are degraded via the fusion of autophagosomes with lysosomes to form autolysosomes. (**B**). Microautophagy, in which cytosolic components are directly engulfed without the formation of vesicles. (**C**). Chaperone-mediated autophagy, in which a selectively targeted protein is attached to a specific KERQ sequence, recognized by Hsc70, and transported to the lumen of the lysosome by LAMP-2A. (Template for figures was adapted from Biorender.com).

**Figure 4 cells-13-00447-f004:**
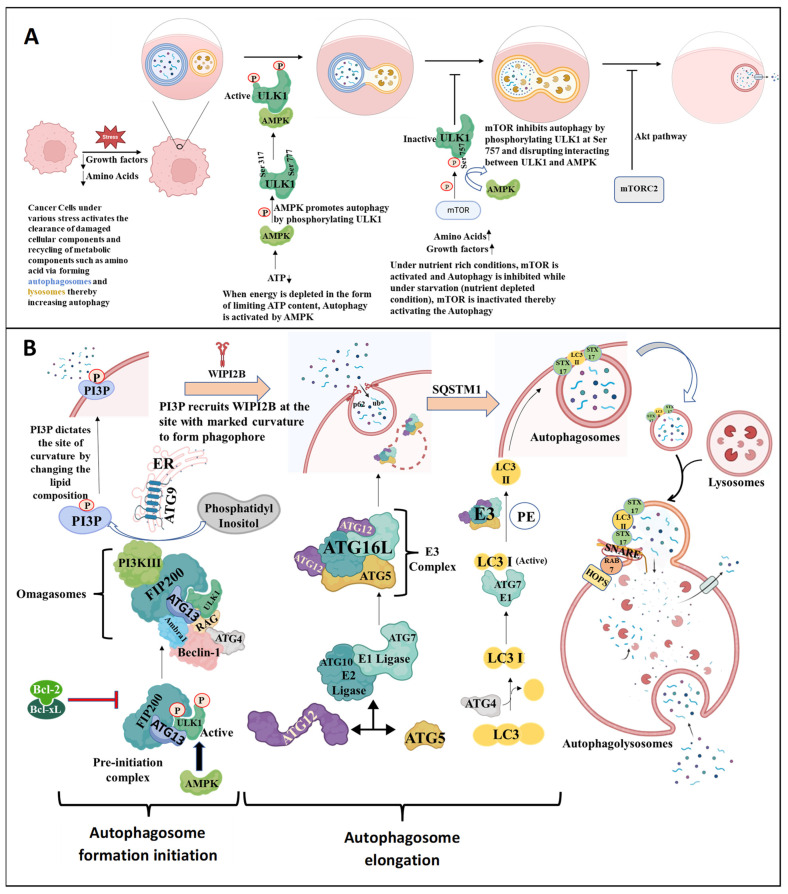
Regulation of autophagy and its mechanisms. (**A**) Regulation—Autophagy is tightly regulated by several key signaling pathways involving mTOR, AMPK, and mTORC2. In conditions of energy depletion, AMPK becomes activated, leading to the phosphorylation of ULK1 at Ser-317 and Ser-777, thereby promoting autophagy. Conversely, when nutrients are abundant, mTOR inhibits autophagy by phosphorylating ULK1 at the Ser-757 residue, which hinders the interaction between ULK1 and AMPK. (**B**) Mechanism of regulation—The process of autophagosome formation begins with the activation of ULK1, which forms a preinitiation complex comprising ATG13 and FIP200. This complex subsequently recruits PI3KIII, initiating the formation of the Omegasome complex, which includes Beclin-1, UV-RAG, ATG14, and Ambra1 at the endoplasmic reticulum (ER), facilitated by ATG9. PI3KIII catalyzes the conversion of phosphatidylinositol to phosphatidylinositol-3-phosphate (PI3P), altering the lipid composition of the membrane. PI3P then recruits WIPI2B, facilitating autophagosome elongation. Autophagosome elongation involves two distinct ubiquitin-like conjugation pathways. ATG12 and ATG5 form a complex with the E1 ligase ATG7 and the E2 ligase ATG10, leading to the formation of the E3 complex. This E3 complex associates with the outer membrane of the phagophore. LC3-I, generated by the cleavage of LC3 by ATG4, is then conjugated to phosphatidylethanolamine (PE) to form LC3-II, a process mediated by the E1 ligase ATG7 and the E3 complex. LC3-II is recruited to both the outer and inner membranes of the autophagosome. As autophagy progresses, the autophagosome membrane expands, engulfing cytoplasmic contents, including proteins tagged with ubiquitin, such as sequestosome1 (SQSTM1)/p62, which bind to LC3. The autophagosome closes, facilitated by STX17, detaching from the ER membrane. This allows the autophagosome to move closer to lysosomes, where fusion occurs, mediated by the HOPS (homotypic fusion and protein sorting) complex. RAB7 adaptors facilitate fusion by binding Q-SNARE with STX17. The fusion of the autophagosome and lysosome forms the autophagolysosome complex, where lysosomal degradation of the contents occurs, releasing degraded material back into the cytoplasm. (The template for figures was adapted from Biorender.com).

**Figure 5 cells-13-00447-f005:**
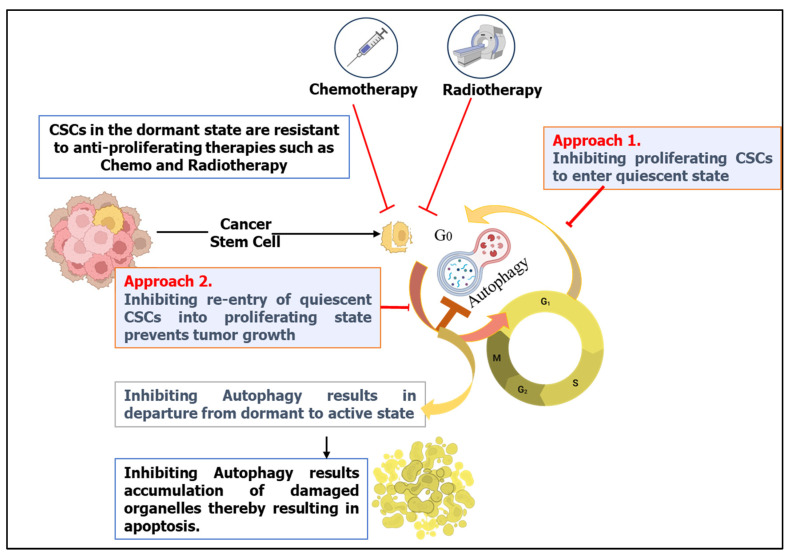
Targeting tumor dormancy through autophagy modulation for therapeutic potential. Tumor relapse can be addressed by targeting tumor dormancy, which can be approached in two ways: the first approach is by inhibiting proliferating CSCs from entering into the quiescent state; another approach is by inhibiting the re-entry of quiescent CSCs into the proliferating state of the cell cycle. Autophagy plays a crucial role in quiescent (G0) CSCs by providing basic requirements for CSC survival, such as amino acids, ATP production, and preventing energetic catastrophe/apoptotic escape. Thus, targeting autophagy can sensitize dormant tumors to mitogens and chemotherapy and address tumor relapse. (Template for figures was adapted from Biorender.com).

**Figure 6 cells-13-00447-f006:**
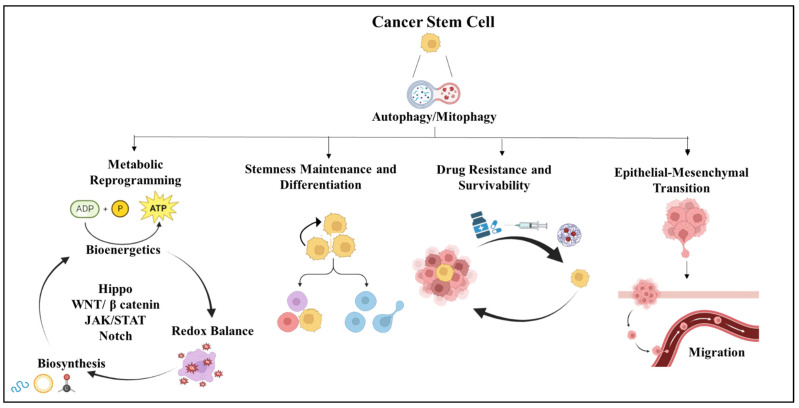
Autophagy in the regulation of cancer stem cells and its therapeutic potential. Cancer stem cells maintain a higher level of autophagy, which plays a role in metabolic reprograming and regulates stemness, differentiation, drug resistance, survival, and epithelial–mesenchymal transition. Metabolic reprograming mediated by autophagy also controls various properties of CSCs, such as epithelial–mesenchymal transition, which promotes migration, drug resistance, stemness, and differentiation. (The template for the figures was adapted from Biorender.com).

## Data Availability

Not applicable.

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
