# Peer review of "Emerging Role of Autophagy in Governing Cellular Dormancy, Metabolic Functions, and Therapeutic Responses of Cancer Stem Cells"

_cells, 2024, doi:10.3390/cells13050447_

Round 1

Reviewer 1 Report (New Reviewer)

Comments and Suggestions for Authors

This is a very well written and overspanning review of Tiwari et al., on the role of autophagy in câncer stem cells.  The writing is cutting edge and the thematic very importante andu p to date.

Having said that a couple of smaller improvements in writing style and figure presentation are required to fully recommend publication:

1.      Page 2, introduction: line 56/57. The logic and semantic conection between the two phrases is ambíguos and misleading: the first frase implies a AML hierarchy model, the second frase continues to explain that “such a population” was also identified.... 

The problem is that this may suggest that the hieracht per se is a poulation... please improve sentences to improve clearness of thought

2.      Figure 1, panel A : title line in the figure contains scrambled letter and numbers in the beginning in the Upper left corner.

3.      All figures versus text: whereas thext is prolific and specific in rich molecular detail, not all figures reflect that and miss many molecular players.

A specific example for clarity: figure 4: in the text ATG13/FIP200/ATG9 are mentioned in connecton with Ulk1, but the do not appear in the figure.

Ideally all that is imortant and mentioned in the text should also be visible and represented in the figures.

Author Response

Response to Reviewer 1

Thanks for the positive response and suggestions to improve the manuscript. Following is a pointwise response.

Comment 1.      Page 2, introduction: line 56/57. The logic and semantic connection between the two phrases is ambiguous and misleading: the first phrase implies a AML hierarchy model, the second phrase continues to explain that “such a population” was also identified.... 

The problem is that this may suggest that the hierarchy per se is a population... please improve sentences to improve clearness of thought

Response: Thanks for the suggestion. We have corrected the sentence.

Comment 2.      Figure 1, panel A : title line in the figure contains scrambled letter and numbers in the beginning in the Upper left corner.

Response: It might be an error due to the pdf. In the word file or our version, we could not see any such mistake.

Comment 3.      All figures versus text: whereas th text is prolific and specific in rich molecular detail, not all figures reflect that and miss many molecular players. 

A specific example for clarity: figure 4: in the text ATG13/FIP200/ATG9 are mentioned in connecton with Ulk1, but they do not appear in the figure.

 Ideally all that is important and mentioned in the text should also be visible and represented in the figures.

Response: Suggestion very well taken, we have replaced the figure 4 and now have provided the details as suggested.

Reviewer 2 Report (New Reviewer)

Comments and Suggestions for Authors

The article by Tiwari et al provides a comprehensive review of the role of autophagy and cancer stem cells (CSCs). The outlining of each section is well organized and each section contains a good overview of the subject. The reviewer recommends the article for publication after the revision.

Line number

100-101 Because of … genetic profiles – sentence unnecessary

112-113 Epithelial … breast cells - sentence seems out of place

119 these models – Only B and C, not A

129 Figure label “B.” is missing

164 proteinsin-proteins in

164-166 The expression … in these cells – sentence seems out of place

171 period missing

191 Reference #43 seems misplaced. Check if this is a correct one

268-269 More detail in figure would be great

272-276 Subsequently… By Bcl-2/Bcl-xL – sentence not clear. Rewrite

285 phaeophore – phagopore?

291 ATG4and 2 – ATG4 and 2

303 autophagosomes move closure to lysosomes – not sure what this means. Rewrite

335 beclin1 comes out of nowhere. Need more background

351 as one of the multiple biomarkers for the same – not sure what this means. Revise

356 co-localization – Co-localization

392 because CSCs are derived from these cells – is this definitive? Wasn’t this just one of the theories?

432 SHH – define SHH. Sonic Hedgehog?

438 IM – define IM, Imatinib Mesylate?

457 mitophage – I think the concept of mitophage has not been dealt at this point. I understand it comes later in the manuscript. Maybe put some short explanations like ‘the details of the concept will be discussed in the consecutive section’

457-459 Sentence too long (two pronoun used in a single sentence) – rewrite

554 Period missing after Ref#9

555 metabolically – metabolic

556 stem traits – stem cell traits

561-563 Two ‘Thus’ used continuously – rewrite

665 trigger the quiescent state – trigger to “enter” the quiescent state?

666 imatinib – isn’t this used as an IM already? Same problems appear throughout the manuscript. Abbreviations used first without defining. Please revise all abbreviations used.

672 Reference missing

674 Define LT-HSCs

713 These metabolic … , in CSCs. – In CSCs, these metabolic

715 report that suggest that – rewrite

722 adenosine 5’-triphosphate and nicotinamide adenine dinucleotide – ATP and NAD has been already used multiple times. Please put the definition when these first appear in the paper

766-767 that driver PDAC – unclear what this means. Rewrite

782-796 the title of the section is about metabolic reprograming. However, it doesn’t seem to contain any content about the metabolic reprograming. Please revise.  

817 – HIF-1a already used as abbreviation without defining. 174, 503… revise

852, 855 these findings appear twice in a row. Rewrite

855-856 These findings … to chemoresistance – sentence unclear. Rewrite

858 TMZ – define TMZ

940-945 There is already a redundant expression in 684. Revise

986, 987 two ‘along with’ used in a single sentence. Rewrite

989 radiotherapy radiation – radiotherapy treatment

1021 two periods

2037 RT – define RT

Figure 6. I don’t think authors dealt with migration in the manuscript. Revise.

Author Response

Response to Reviewer 2.

We are thankful to the reviewer for your valuable comments and suggestions. We appreciate the thorough review and have carefully implemented all the suggested changes. The corrections have been highlighted in yellow throughout the manuscript, and corresponding responses have been provided in the comment box. Here is a brief summary of our responses to each point:

Comment: 100-101 Because of … genetic profiles – sentence unnecessary

Response: Correction done

Comment:112-113 Epithelial … breast cells - sentence seems out of place

Response: Correction done

Comment:119 these models – Only B and C, not A

Response: Correction done

Comment:129 Figure label “B.” is missing

Response: Correction done

Comment:164 proteinsin-proteins in

Response: Correction done

Comment:164-166 The expression … in these cells – sentence seems out of place

Response: Correction done

Comment:171 period missing

Response: Correction done

Comment:191 Reference #43 seems misplaced. Check if this is a correct one

Response: Correction done

Comment:268-269 More detail in figure would be great

Response: Correction done

Comment:272-276 Subsequently… By Bcl-2/Bcl-xL – sentence not clear. Rewrite

Response: Correction done

Comment:285 phaeophore – phagopore?

Response: Correction done

Comment:291 ATG4and 2 – ATG4 and 2

Response: Correction done

Comment: 303 autophagosomes move closure to lysosomes – not sure what this means. Rewrite

Response: Correction done

Comment: 335 beclin1 comes out of nowhere. Need more background

Response: Correction done

Comment: 351 as one of the multiple biomarkers for the same – not sure what this means. Revise

Response: Correction done

Comment: 356 co-localization – Co-localization

Response: Correction done

Comment: 392 because CSCs are derived from these cells – is this definitive? Wasn’t this just one of the theories?

Response: Correction done

Comment: 432 SHH – define SHH. Sonic Hedgehog?

Response: Correction done

Comment: 438 IM – define IM, Imatinib Mesylate?

Response: Correction done

Comment: 457 mitophage – I think the concept of mitophage has not been dealt at this point. I understand it comes later in the manuscript. Maybe put some short explanations like ‘the details of the concept will be discussed in the consecutive section’

Response: Correction done

Comment: 457-459 Sentence too long (two pronoun used in a single sentence) – rewrite

Response: Correction done

Comment: 554 Period missing after Ref#9

Response: Correction done

Comment: 555 metabolically – metabolic

Response: Correction done

Comment: 556 stem traits – stem cell traits

Response: Correction done

Comment: 561-563 Two ‘Thus’ used continuously – rewrite

Response: Correction done

Comment: 665 trigger the quiescent state – trigger to “enter” the quiescent state?

Response: Correction done

Comment: 666 imatinib – isn’t this used as an IM already? Same problems appear throughout the manuscript. Abbreviations used first without defining. Please revise all abbreviations used.

Response: Correction done

Comment: 672 Reference missing

Response: Correction done

Comment: 674 Define LT-HSCs

Response: Correction done

Comment: 713 These metabolic … , in CSCs. – In CSCs, these metabolic

Response: Correction done

Comment: 715 report that suggest that – rewrite

Response: Correction done, the sentence is rewritten.

Comment: 722 adenosine 5’-triphosphate and nicotinamide adenine dinucleotide – ATP and NAD has been already used multiple times. Please put the definition when these first appear in the paper

Response: Correction done

Comment: 766-767 that driver PDAC – unclear what this means. Rewrite

Response: Correction done, the sentence is rewritten

Comment: 782-796 the title of the section is about metabolic reprograming. However, it doesn’t seem to contain any content about the metabolic reprograming. Please revise.  

Response: Correction done. Revised.

Comment: 817 – HIF-1a already used as abbreviation without defining. 174, 503… revise

Response: Correction done. Revised.

Comment: 852, 855 these findings appear twice in a row. Rewrite

Response: Correction done, the sentence is rewritten

Comment: 855-856 These findings … to chemoresistance – sentence unclear. Rewrite

Response: Correction done

Comment:858 TMZ – define TMZ

Response: Correction done

Comment: 940-945 There is already a redundant expression in 684. Revise

Response: Correction done

Comment:986, 987 two ‘along with’ used in a single sentence. Rewrite

Response: Correction done

Comment: 989 radiotherapy radiation – radiotherapy treatment

Response: Correction done

Comment: 1021 two periods

Response: Correction done

Comment: 2037 RT – define RT

Response: Correction done

Comment: Figure 6. I don’t think authors dealt with migration in the manuscript. Revise.

Response: Thanks for pointing it out. We have corrected it (we mentioned Epithelial to mesenchymal transition that leads to migration)

This manuscript is a resubmission of an earlier submission. The following is a list of the peer review reports and author responses from that submission.

Round 1

Reviewer 1 Report

Comments and Suggestions for Authors

The authors attempt to summarize the knowledge about autophagy, describing its forms, causes, regulation and effects. For the most part is a solid review, well documented, current and likely to garner interest. There are, however a few comments to be made.

"Tumor heterogeneity poses a biggest challenge for the clinical management of cancer due to lack of effective treatment regime."

I honestly have never seen this affirmation anywhere else. Generally speaking, oncologists agree that formation of metastatic foci is the most prominent problem faced in clinic and the main cause of mortality among cancer patients, followed by resistance phenotypes. As this is not repeated anywhere else throughout the text, there is no citation to confirm this. You can rewrite it, give it more context in the main text or scrape it altogether.

It would be interesting to add a small section mentioning the role glycoconjugates play in the induction of autophagy as aberrant glycosylation patterns are a hallmark of cancer cells. For instance, lectins binding sialic acid are capable of inducing autophagy. Also, gangliosides are constituents of the autophagosome membrane and glycan signaling from the extracellular matrix impacts the expression and localization of autophagy regulators, such as AMPK and mTORC1.

Comments on the Quality of English Language

The manuscript is understandable due to a combination of context and knowledge of the subject, but it's currently not a pleasant nor an easy read.

Author Response

Response to the Reviewer 1:

We are really thankful to the reviewer for the suggestions. We have incorporated the comments and corrections are done accordingly in the review. Pointwise reply to the comments is provide in the following sections.

Comments:

"Tumor heterogeneity poses a biggest challenge for the clinical management of cancer due to lack of effective treatment regime." I honestly have never seen this affirmation anywhere else. Generally speaking, oncologists agree that formation of metastatic foci is the most prominent problem faced in clinic and the main cause of mortality among cancer patients, followed by resistance phenotypes. As this is not repeated anywhere else throughout the text, there is no citation to confirm this. You can rewrite it, give it more context in the main text or scrape it altogether.

Response: We really appreciate the comments provided by the reviewer. We agree with the comment and we have reframed the sentence accordingly (in the abstract sections, page no. 1).

Comment: It would be interesting to add a small section mentioning the role glycoconjugates play in the induction of autophagy as aberrant glycosylation patterns are a hallmark of cancer cells. For instance, lectins binding sialic acid are capable of inducing autophagy. Also, gangliosides are constituents of the autophagosome membrane and glycan signaling from the extracellular matrix impacts the expression and localization of autophagy regulators, such as AMPK and mTORC1.

Response: The point is well taken and we do agree that the glycoconjugates are an important link between metastasis and autophagy. We have incorporated paragraphs discussing the link between autophagy and glycoconjugates in cancer and CSCs. (in subsection 3.2. Autophagy in Cancer: Complex Role, page no. 8-9)

Reviewer 2 Report

Comments and Suggestions for Authors

The manuscript provides a comprehensive and insightful overview of the emerging role of autophagy in governing cellular dormancy, metabolic functions, and therapeutic responses of cancer stem cells. The authors have done an excellent job synthesizing the existing literature and presenting a clear and coherent narrative. The manuscript is also well-organized, with each section building upon the previous one to provide a thorough understanding of the topic.

Author Response

Reviewer 2:

Comment: The manuscript provides a comprehensive and insightful overview of the emerging role of autophagy in governing cellular dormancy, metabolic functions, and therapeutic responses of cancer stem cells. The authors have done an excellent job synthesizing the existing literature and presenting a clear and coherent narrative. The manuscript is also well-organized, with each section building upon the previous one to provide a thorough understanding of the topic.

Response: We are really thankful to the reviewer for appreciating the review. It is really encouraging.

Reviewer 3 Report

Comments and Suggestions for Authors

The Tiwari manuscript might be interesting, but it is not acceptable in its present form. It is badly and loosely written. There are grammatical errors, repetitions (e.g., “inhibition of mitophagy via inhibition by DRP1 inhibits the reprogramming” line 476), incorrect semicolon (; which), dash or space. The structure and subdivision of the manuscript is approximate (3.4; 4.3 and 4.4;5.1. Quiescence of CSCs and 5.3. Quiescent State in CSCs). The manuscript contains contents repeated many times, others unclear or missing. Even the figures are not clear (e.g., figure 4, 5) or not very useful (figure 2); legend 4 contains an error. Some references are wrong (e.g., 127- 132-133). Extremely important aspect the abstract of reference 126 was copied verbatim from lines 535 to 544.

Comments on the Quality of English Language

The Tiwari manuscript might be interesting, but it is not acceptable in its present form. It is badly and loosely written. There are grammatical errors, repetitions (e.g., “inhibition of mitophagy via inhibition by DRP1 inhibits the reprogramming” line 476), incorrect semicolon (; which), dash or space. The structure and subdivision of the manuscript is approximate (3.4; 4.3 and 4.4;5.1. Quiescence of CSCs and 5.3. Quiescent State in CSCs). The manuscript contains contents repeated many times, others unclear or missing. Even the figures are not clear (e.g., figure 4, 5) or not very useful (figure 2); legend 4 contains an error. Some references are wrong (e.g., 127- 132-133). Extremely important aspect the abstract of reference 126 was copied verbatim from lines 535 to 544.

Author Response

Response to Reviewer 3: 

We are really thankful to the reviewer for taking out time and providing constructive critical comments to improve the review. We have taken in account all the suggestions made and corrected the manuscript accordingly. Following is the pointwise reply to the comments.

Comment: It is badly and loosely written. There are grammatical errors, repetitions (e.g., “inhibition of mitophagy via inhibition by DRP1 inhibits the reprogramming” line 476), incorrect semicolon (; which), dash or space.

Response: We have carefully corrected the review and apologies for such errors while submitting the review. At the same time we thank the reviewer for time and efforts in pointing out such errors. It has helped us immensely to improve the review.

Comment: The structure and subdivision of the manuscript is approximate (3.4; 4.3 and 4.4;5.1. Quiescence of CSCs and 5.3. Quiescent State in CSCs).

Response: We do agree with the comment, but at the same time, there are topic that are extensively studied or not so extensively studied, thus are subdivided accordingly for better understanding. However, we have done corrections wherever possible.

Comment: The manuscript contains contents repeated many times, others unclear or missing. Even the figures are not clear (e.g., figure 4, 5) or not very useful (figure 2); legend 4 contains an error.

Response: We have improved the figures as per the suggestions. Error in figure legend 4 has been corrected.

Comment: Some references are wrong (e.g., 127- 132-133). Extremely important aspect the abstract of reference 126 was copied verbatim from lines 535 to 544.

Response: Thanks for pointing out these error, the corrections have been made accordingly.

Round 2

Reviewer 1 Report

Comments and Suggestions for Authors

The current version of the manuscript is a great improvement on the original draft. The changes made by the authors turned the review into a much more sound, comprehensive and useful text. 

Comments on the Quality of English Language

English language was also greatly improved, but there are still minor issues. The very first sentence contains repeated words. "The tumour is composed tumor of heterogeneous population..." As is it literally the first sentence of the abstract it is a bit of an eye sore.

There are other minor grammar and punctuation mistakes throughout the text that should be addressed.  

As a minor note, the authors changed every instance of tumour to tumor. Both are right, but the first is used in the UK and the former in the US. There are many other words that are spelled differently. One example is leukemia, which is also used in the manuscript. In the UK it is spelled leukaemia. It would be better to choose either UK or US English and stick with it.

Author Response

Response to the reviewer.

Comments: The current version of the manuscript is a great improvement on the original draft. The changes made by the authors turned the review into a much more sound, comprehensive and useful text. 

Response: Thanks for the positive comments. We have further improved the manuscript by using professional editing.

Comments on the Quality of English Language

English language was also greatly improved, but there are still minor issues. The very first sentence contains repeated words. "The tumour is composed tumor of heterogeneous population..." As is it literally the first sentence of the abstract it is a bit of an eye sore.

There are other minor grammar and punctuation mistakes throughout the text that should be addressed.  

As a minor note, the authors changed every instance of tumour to tumor. Both are right, but the first is used in the UK and the former in the US. There are many other words that are spelled differently. One example is leukemia, which is also used in the manuscript. In the UK it is spelled leukaemia. It would be better to choose either UK or US English and stick with it.

Response: We are thankful for the comments. Now we have used a language editing software (professional) and corrected the manuscript accordingly.
